## PROCEEDINGS A

computational mechanics, structural engineering, mechanics

buckling, localization, snaking, pattern formation

**Author for correspondence:**
R. M. J. Groh
e-mail: rainer.groh@bristol.ac.uk

# On the role of localizations in buckling of axially compressed cylinders

## R. M. J. Groh and A. Pirrera

Department of Aerospace Engineering, Bristol Composites Institute (ACCIS), University of Bristol, BS8 1TR Bristol, UK

RMJG, 0000-0001-5031-7493

The collapse of axially compressed cylinders by buckling instability is a classic problem in engineering mechanics. We revisit the problem by considering fully localized post-buckling states in the form of one or multiple dimples. Using nonlinear finite-element methods and numerical continuation algorithms, we trace the evolution of odd and even dimples into one axially localized ring of circumferentially periodic diamond-shaped waves. The growth of the post-buckling pattern with varying compression is driven by homoclinic snaking with even- and odd-dimple solutions intertwined. When the axially localized ring of diamond-shaped buckles destabilizes, additional circumferential snaking sequences ensue that lead to the Yoshimura buckling pattern. The unstable single-dimple state is a mountain-pass point in the energy landscape and therefore forms the smallest energy barrier between the pre-buckling and post-buckling regimes. The small energy barrier associated with the mountain-pass point means that the compressed, pre-buckled cylinder is exceedingly sensitive to perturbations once the mountain-pass point exists. We parameterize the compressive onset of the single-dimple mountain-pass point with a single non-dimensional parameter, and compare the lower-bound buckling load suggested by this parameter with over 100 experimental data points from the literature. Good correlation suggests that the derived knockdown factor provides a less conservative design load than NASA's SP-8007 guideline.

# 1. Introduction

Thin-walled shell structures are widely employed as mass-efficient means to carry loads. Their occurrence is ubiquitous, both in nature (e.g. egg shells) and in human-made constructions (e.g. rocket launchers, ship hulls, cars, roofs, etc.). Using shell structures safely in engineering applications requires the ability to predict their failure loads accurately and systematically.

Due to their slenderness, thin-walled structures are particularly prone to buckling instabilities when subject to compressive loads. When the pre-buckling equilibrium path is linear, predictions of the first instability load can be based on a linearized eigenproblem. This approach provides excellent correlations between theory and experiments for thin-walled beams and flat plates. However, this is generally not the case for curved shell structures.

For a thin walled, isotropic cylinder of Young's Modulus $E$, Poisson's ratio $v$, thickness $t$ and simply supported edges, a classic linear eigenproblem based on small-deflection theory [1] suggests a compressive buckling stress of:

$$\sigma_{cl} = \frac{E}{\sqrt{3(1 - v^2)}} \frac{t}{R}. \tag{1.1}$$

Experimental tests, on the other hand, can show collapse loads as low as 20% of this value. The challenge of closing the gap between theoretical predictions and experimental measurements has drawn the attention of structural mechanicians for better parts of the twentieth century. Today, developments in computational mechanics and experimental techniques facilitate new approaches and provide new perspectives on this classical problem. With a renewed interest in space exploration and evolving material systems, the opportunity exists to design a new generation of efficient cylindrical launch structures, and by using contemporary insights, to update legacy design guidelines. The present work fits within this context.

As the buckling of a thin walled, elastic cylinder is governed by a subcritical pitchfork bifurcation (figure 1*a*), the post-buckling equilibrium path is initially unstable and falls rapidly from the classical buckling load $\sigma_{cl}$. In the early part of the twentieth century, von Kármán & Tsien [2] showed that this large deformation, post-buckled equilibrium path restabilizes at a lower point, $\sigma_L \ll \sigma_{cl}$. Von Kármán & Tsien went on to suggest that this minimum load could serve as a useful conservative buckling load of the inevitably imperfect shell. The next breakthrough came with Koiter's general theory of elastic stability [3], in which an asymptotic perturbation approach was used to quantify the sensitivity of the idealized buckling load with respect to small geometric imperfections.

Indeed, the sharp subcritical nature of the bifurcation makes the idealized cylinder extremely sensitive to initial imperfections; imperfections that round-off the idealized bifurcation point into a series of limit points of rapidly decreasing load for increasing imperfection amplitude. While imperfections can take many forms—from variations in shell thickness and material properties, to non-uniform loading and boundary conditions—the greatest factor contributing to the decrease in buckling load can be related to the influence of geometric imperfections of the cylinder's mid-surface [5].

One of the limitations of Koiter's perturbation approach is that it is based on expansions of the energy close to the bifurcation point, limiting its applicability to small imperfections. For example, Zhu *et al.* [6] note that classical theories fail to account for the nonlinear relationship between buckling stress ($\sigma_{cl}$) and thickness-to-radius ratio ($t/R$). While classical theory (see equation (1.1)) suggests a buckling stress proportional to $t/R$, empirical data show the stress to be proportional to $(t/R)^{1.5}$. Zhu *et al.* [6] argue that this discrepancy can be explained by analysing inherently imperfect shells, rather than idealized ones with small imperfections. Indeed, a contemporary revival of interest in shell buckling has focused on conducting fully nonlinear post-buckling analyses—often with initially applied imperfections—in an attempt to gain further insight into the collapse of shells from the plethora of non-trivial and entangled post-buckling equilibrium paths [7]. With modern rapid-prototyping techniques, shells with

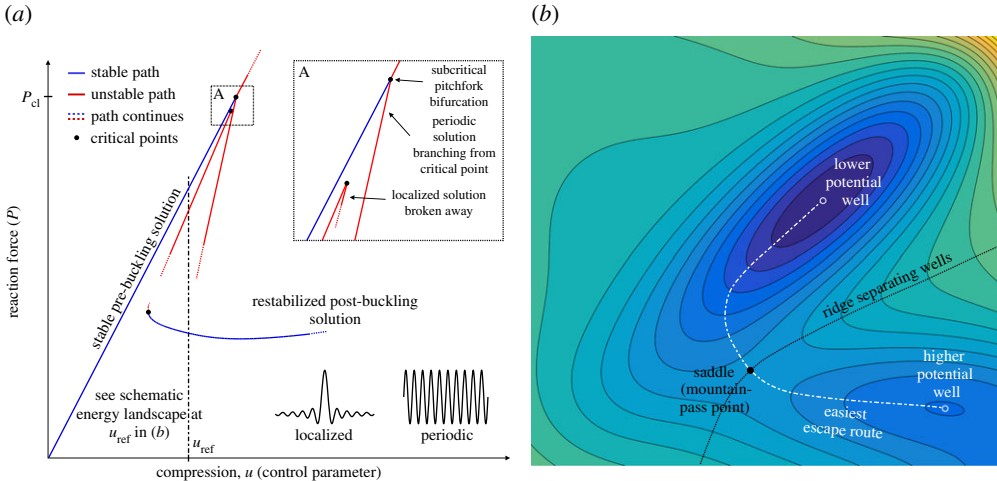

**Figure 1.** (*a*) Schematic equilibrium diagram (reaction force *P* versus applied compression *u*) of an axially compressed cylinder featuring a linearly stable pre-buckling equilibrium path that destabilizes at a subcritical pitchfork bifurcation. Apart from the unstable periodic post-buckling path that branches from the critical point, broken-away localized equilibrium paths also exist (one is shown here for illustration but more are shown in figures 3 and 4) that are delimited by turning points in the immediate vicinity of the pitchfork bifurcation. The post-buckling regime features many different restabilized equilibrium segments (one is shown here) that are of lower energy (bending dominated) than the fundamental pre-buckling equilibrium (membrane dominated). These restabilized segments may be connected to either localized or periodic paths. As shown schematically in the energy landscape of (*b*), for one level of compression ($u_{ref}$), the stable pre-buckling and restabilized post-buckling energy wells are separated by a ridge of unstable hilltops. The bounding ridge, which may comprise a set of intersecting hilltop curves, may feature both periodic and localized states. The lowest point on the bounding ridge (a saddle) is termed a *mountain-pass point* and corresponds to a single dimple in the cylinder wall [4]. A single-dimple localization, therefore, provides the easiest route to prematurely escape the pre-buckling well before the first instability point is reached. (Online version in colour.)

precisely engineered imperfections can be manufactured, and recent studies have confirmed that excellent correlation between experiments and predictions can be achieved, if the precise nature of imperfections is incorporated during modelling [8]. Possible shortcomings of this approach in engineering practice are that accurate imperfection measurements are expensive to come by, and can vary from specimen to specimen depending on the chosen material system and manufacturing method. Furthermore, during the preliminary design and optimization phases, geometric and material imperfections, as well as the precise nature of the loading conditions, are not known in detail. As a consequence, structural analysts often rely on lower-bound (safe) estimates of the buckling load, e.g. NASA's empirical SP-8007 guideline [9]. Knowledge of the 'worst' type of geometric imperfection, and its likelihood, thus remain useful during the early design stages.

Following Koiter [10], geometric imperfections based on the eigenvector of the first bifurcation point of the perfect cylinder are considered to be the most conducive to buckling. For example, imperfections based on axisymmetric eigenmodes can lead to collapse at 30% of $\sigma_{cl}$, for imperfection amplitudes of only one-half the wall thickness [10]. Because this particular (structured) imperfection shape is not likely in engineering practice, many imperfection sensitivity analyses use imperfections that are linear combinations of the first $n_{cl}$ eigenmodes [11]. One shortcoming of this approach is that localized imperfections, such as one or multiple dimples, are difficult to represent as linear combinations of the fundamentally periodic eigenmodes; yet localizations play a key role in the initiation of buckling in axially compressed cylinders [12,13]. As noted in the high-speed photography study by Eßlinger & Geier: 'for short or very thin cylinders local buckles always occur and a circumferentially evenly distributed post-buckling

waveform can hardly be induced [...] also for long, less thin-walled cylinders, which show a periodic post-buckling pattern after buckling, the buckling process begins locally with the formation of a single buckle' [12, translated from German by the first author]. These observations led Eßlinger & Geier to formulate the hypothesis that a lower-bound buckling load could be based on stress concentrations stimulated by local initial undulations or external disturbances. Such non-uniform load distributions had, for example, been recorded immediately before buckling by Babcock & Sechler [14, p. 38–41].

Localized (concentrated) buckling modes differ from periodic eigenmodes in that they are confined to a portion of the structure's geometry (figure 1a). Although interacting periodic eigenmodes can have a modulating effect [15], each 'apparent' localization of superposed eigenmodes corresponds to a separate equilibrium path. On the other hand, as pointed out by Hunt *et al.* [16], the pure form of a localization is necessarily multi-valued, and therefore able to develop with equal likelihood in many places over the spatial domain, each time resulting in the same equilibrium path. A localized equilibrium path is therefore only one instance in a large set—and for infinitely 'long' structures, strictly an infinite set—of competing possibilities. Because this situation with potentially infinite alternative equilibrium states has certain analogies with temporal chaos it is sometimes referred to as 'spatial chaos' [17].

Localizations, such as a single dimple, are typical of structures with subcritical (unstable) post-buckling behaviour, appearing either as secondary bifurcations from a periodic post-buckling path or as broken-away equilibrium paths juxtaposed to the pre-buckling path [16,18] (figure 1a). These features make it difficult to identify localized post-buckling states using a linear eigenproblem on the pre-buckling path. Instead, asymptotic approaches that assume both frequency and amplitude modulation [19], or alternatively, numerical continuation algorithms that can path-follow the pertinent localized equilibrium paths, are required.

Using such an asymptotic approach that allows the buckling mode amplitude to modulate axially, Hunt & Lucena Neto [20] showed that post-buckling modes localized along the length of the cylinder exist in the form of a circumferential ring of diamond-shaped waves. Hunt *et al.* [21] then showed that this axially localized ring can undergo homoclinic snaking (also known as cellular buckling) in the fundamental loading parameter, sequentially adding more and more rings along the length of the cylinder through a series of de- and re-stabilizations. A seminal work by Horák *et al.* [4] further indicated that fully localized equilibrium solutions—i.e. localized both in the axial and circumferential directions in the form of one or multiple dimples—also exist. In an experimental setting, tests on axially compressed cylinders have shown that the single dimple (a buckle in the shape of an inwards dent) is indeed an equilibrium state [22]. Kreilos & Schneider [23] showed that such a single-dimple localization can undergo snaking in the circumferential direction until the circumference of the cylinder is filled with one ring of diamond buckles. Once this ring is complete, cellular buckling in the axial direction, as previously suggested by Hunt *et al.* [21], is possible, although the connection between these two phenomena could not be shown. The important role of dimple localizations in the buckling of spherical shells under external pressure has also recently been addressed by Hutchinson [24] and Hutchinson & Thompson [25].

Interestingly, for a specific level of end-shortening, the single dimple identified by Horák *et al.* [4] corresponds to the smallest energy barrier between the pre-buckled state and a restabilized post-buckling state of lower energy. An important caveat is that the cylinder only restabilizes under controlled end-shortening (rigid loading), whereas in the case of force-controlled loading (dead loading), the cylinder undergoes complete collapse once the mountain pass has been crossed. With this caveat in mind, consider, for example, the two energy wells depicted in figure 1b, with one valley representing a high-energy pre-buckling equilibrium and the other a restabilized, low-energy post-buckling equilibrium (of localized or periodic form). These two energy wells are necessarily separated by a ridge of unstable equilibria, with the lowest point on this ridge, i.e. a saddle in the energy landscape, corresponding to the aforementioned minimum energy barrier—a so-called *mountain-pass point*. Because a whole plethora of restabilized post-buckling states of lower energy than the pre-buckling state generally exist [4],

the basin of attraction surrounding the pre-buckling well features multiple mountain-pass points. However, Horák *et al.* [4] have, at least numerically, shown that the single-dimple forms the smallest of all energy barriers, and this is the mountain-pass point we henceforth refer to.

The existence of a mountain-pass point essentially follows from two conditions: (i) the pre-buckled state is a local minimum, and (ii) a post-buckling state of lower energy exists. Thus, the existence of a mountain-pass point implies that the system can, in principle, escape from the pre-buckling state to another equilibrium of lower energy; or, in other words, a mountain-pass point implies that the pre-buckling state is *metastable*. The existence of a mountain-pass point is thus a useful piece of information in estimating the sensitivity of the cylinder to perturbations. To escape the pre-buckling energy well, any perturbation must impart sufficient energy to traverse the barrier provided by the mountain-pass point. The small energy barriers associated with localized mountain-pass states [7], suggest a possible mechanism for premature buckling: the traversal or erosion of energy barriers due to initial imperfections, loading eccentricities, statistical fluctuations or external disturbances, i.e. 'shocks' [26]. Quantifying the level of compression for which such 'shock sensitivity' arises may thus be a fruitful approach for quantifying buckling loads for new design guidelines. Indeed, recent work on the buckling of spherical shells under external pressure suggests that the onset of 'shock sensitivity' can be used as a design load for spherical caps [27].

In this paper, we analyse an axially compressed, isotropic cylinder and trace the post-buckling equilibria of odd and even number of dimples using quasi-static, nonlinear finite-element methods coupled to numerical continuation algorithms. The results provide insights into the formation of the Yoshimura diamond waveform through a sequence of homoclinic snaking phenomena. In addition, we use the compressive onset of the single dimple as a mountain-pass point to formulate buckling estimates that are less conservative than NASA's SP-8007 guideline. Thus, rather than analysing imperfect cylinders, we derive a measure of the perturbation sensitivity from the energy landscape of the perfect cylinder.

The rest of the paper is structured as follows. Section 2 provides a brief overview of the finite-element model and the numerical continuation framework. The pre-buckling and periodic post-buckling equilibrium curves of the cylinder are presented in §3, which are followed in §4 by equilibrium manifolds depicting the circumferential growth of localized post-buckling equilibria into one ring of circumferential diamonds and further into the Yoshimura pattern. The stability landscape and associated energy barriers are assessed in §5. The relation between metastability, the existence of a mountain-pass point, and the Maxwell energy criterion is discussed in §6. Finally, a new design load based on the compressive onset of the single dimple as a mountain-pass point is introduced in §7, and conclusions are drawn in §8.

## 2. Computational model

We consider a thin-walled cylindrical shell of thickness $t = 0.247$ mm, radius $R = 100$ mm and length $L = 160.9$ mm loaded by uniform axial compression via displacement control (rigid loading). The cylinder is assumed to be linear elastic and isotropic with Young's modulus $E = 5.56$ GPa and Poisson's ratio $\nu = 0.3$. The particular geometric and material parameters are those of Yamaki's longest cylinder (Batdorf parameter $Z = L^2\sqrt{1 - \nu^2}/Rt = 1000$) [28]. To represent a typical experimental set-up as closely as possible, the cylinder is rigidly clamped at both ends with axial compression permitted at one end but constrained at the other. The displacement-controlled loading is introduced by moving all points on the circumference in the direction of the cylinder axis.

The cylinder is discretized using isoparametric, geometrically nonlinear finite-elements based on a total Lagrangian formulation, i.e. the conjugate strain and stress measures are the Green–Lagrange strain tensor and the second Piola–Kirchhoff stress tensor, respectively, with all tensor quantities defined in a reference frame of the undeformed configuration. The finite elements used are so-called 'degenerated shell elements' [29] based on the assumptions of first-order shear

deformation theory [30] (shear correction factor $k = 5/6$). The cylinder is thus modelled by shell elements with three nodal displacements in the global Cartesian system, and two nodal rotations that describe the finite rotations of the shell directors at the nodes (no drilling around the shell director allowed). Many different ways of parameterizing the rotations of the shell director are possible [31]. Here, we define all incremental nodal rotations with respect to a nodal convected coordinate system lying in the plane of each element [32].

To reduce the computational effort and complexity of the problem, only a quarter of the cylinder is modelled with the pertinent mirror symmetry conditions applied at the cylinder half-length and half-circumference. The imposed translational and rotational symmetry conditions prevent movement of boundary nodes across the symmetry plane and constrain rotations around the line of symmetry. Although symmetry conditions constrain the set of symmetry-breaking bifurcations that can be observed, the results in the following sections show that the observed behaviour is sufficiently rich to warrant this simplification. Future work will extend the model to the full cylinder. The quarter-cylinder is discretized into a mesh of 52 axial and 121 circumferential nodes to form a mesh of 680 fully integrated, 16-noded shell elements with a total of 31 460 d.f. The effects of shear and membrane locking are minimized by using bi-cubic isoparametric interpolation functions (16-noded elements). The mesh was refined until the first 20 bifurcation points while path-following along the pre-buckling path converged to 0.1% of the values obtained from the first 20 eigenvalues of a linear buckling analysis conducted in the commercial finite-element software ABAQUS with a mesh of $300 \times 300$ S4R elements. Furthermore, refinements in the mesh did not qualitatively or quantitatively change the observed mechanical behaviour described herein, and where comparisons are possible, the results match well with previous work by Kreilos & Schneider [23].

In the applied mathematics community, the methods of nonlinear multi-parameter analysis, branch switching and critical point tracking are well established [33]. In engineering mechanics, specialized arc-length solvers restricted to a single parameter (an applied load or displacement) are predominantly used [34]. Apart from tracing one type of load–displacement curve, broader numerical continuation methods coupled to implicit, quasi-static finite-element solvers allow the analyst to distinguish between stable and unstable equilibria; pinpoint critical points, e.g. limit and bifurcation points; branch-switch to additional equilibrium paths at bifurcation points; and efficiently trace critical points in parameter space.

In a classical displacement-based finite-element setting, an equilibrium state is expressed as a balance between internal forces, $f$, and external forces, $p$,

$$F(u, \lambda) = f(u) - p(\lambda) = 0, \tag{2.1}$$

with $n$ displacement degrees-of-freedom, $u$, and a scalar loading parameter, $\lambda$. In a generalized setting, equation (2.1) is adapted to incorporate any number of additional parameters:

$$F(u, \Lambda) = f(u, \Lambda_1) - p(\Lambda_2) = 0, \tag{2.2}$$

where $\Lambda = [\Lambda_1^\top, \Lambda_2^\top]^\top = [\lambda_1, \dots, \lambda_p]^\top$ is a vector containing $p$ control variables.

The expression in equation (2.2) describes $n$ equilibrium equations in $n$ displacement degrees of freedom. Because the system is parameterized by $p$ additional parameters, a $p$-dimensional solution manifold in $\mathbb{R}^{(n+p)}$ exists. Specific solution subsets on this $p$-dimensional manifold are found by defining bordering equations, $g$:

$$G(u, \Lambda) \equiv \begin{pmatrix} F(u, \Lambda) \\ g(u, \Lambda) \end{pmatrix} = 0. \tag{2.3}$$

In total, $p - 1$ bordering equations are required to define a one-dimensional subset curve on the multi-dimensional solution manifold. These bordering equations can define, among others, fundamental equilibrium paths (the fundamental load parameter is varied); parametric equilibrium paths (a non-load parameter is varied); and paths that track critical points (limit and bifurcation points) in parameter space. To solve for critical points, for example, we need to enforce

a criticality condition, e.g. $F_{,u}\phi = 0$ with $\phi$ a critical eigenvector of the Jacobian $F_{,u}$ corresponding to an eigenvalue $\mu = 0$. In the most general form, a vector of $q$ auxiliary variables, $v$, may be added to the bordering equations $g$,

$$G(u, \Lambda, v) \equiv \begin{pmatrix} F(u, \Lambda) \\ g(u, \Lambda, v) \end{pmatrix} = 0. \qquad (2.4)$$

Following the example for critical point tracking referenced above, a critical subset curve in two parameters, $p = 2$, is appropriately constrained by the associated bordering equations $F_{,u}v = 0$ (criticality) and $||v||_2 = 1$ (normalization constraint).

When evaluating a one-dimensional curve, one additional equation is needed to uniquely constrain the system to a solution point $y = (u, \Lambda, v)$. Hence,

$$G^N(y) \equiv \begin{pmatrix} F(u, \Lambda) \\ g(u, \Lambda, v) \\ N(u, \Lambda) \end{pmatrix} = 0, \qquad (2.5)$$

where $N$ is a scalar equation that plays the role of a multi-dimensional arc-length constraint. A specific solution to equation (2.5) is determined by a consistent linearization coupled with a Newton–Raphson algorithm,

$$y_k^{j+1} = y_k^j - \left( G_{,y}^N(y_k^j) \right)^{-1} \cdot G^N(y_k^j) \equiv y_k^j + \delta y_{k'}^j \qquad (2.6)$$

where the superscript denotes the $j^{th}$ equilibrium iteration and the subscript denotes the $k^{th}$ load increment. Detailed descriptions of the generalized path-following framework used herein can be found in Eriksson [35] and Groh *et al.* [36].

The following sections analyse post-buckling localizations in the axially compressed cylinder using this generalized path-following framework. Hence, nonlinear equilibrium paths are traced; critical points (limit and bifurcation points) are determined when the stability along a path changes; branch-switching and path-following along bifurcated paths is conducted when required; and critical boundaries are traced in parameter space. In all equilibrium manifolds that follow, blue segments denote stable equilibria, red segments unstable equilibria and black dots denote critical points.

## 3. Pre-buckling and periodic post-buckling behaviour

The stable pre-buckling behaviour of the axially compressed cylinder is depicted in figure 2a. The metrics used to depict the behaviour in this and all other equilibrium manifolds (unless stated otherwise) are the normalized compressive displacement $uR/Lt$ (independent parameter $u$) versus the normalized reaction force $P/P_{cl}$ on the actuated edge. $P_{cl}$ is the classical buckling load derived from equation (1.1): $P_{cl} = 2\pi Et^2/\sqrt{3(1 - v^2)}$.

The fundamental path loses stability at a symmetry-breaking pitchfork bifurcation with a normalized reaction load of $P/P_{cl} = 0.923$, which correlates closely with the theoretical predictions by Yamaki [28, pp. 233, 253] based on Donnell-von Kármán equations and accounting for clamped edges ($P/P_{cl} \approx 0.92$). The 8% reduction in load compared to the classical prediction is due to the boundary layer induced by the clamped edges, which is not captured by the model underlying equation (1.1). The mode shape of the normalized radial (out-of-plane) displacement, $w/t$, just before buckling (point 1 on the equilibrium path), is illustrated in figure 2b, and clearly shows the boundary layer. All deformation plots shown herein are produced by taking the displacement field of the quarter-model and applying the appropriate reflections along the axial and circumferential directions.

Figure 2a also shows the periodic post-buckling path that branches from the first symmetry-breaking bifurcation. The equilibrium curve is unstable and doubles back close to the fundamental path, confirming the sharp subcritical nature of shell buckling. The mode shapes on this periodic post-buckling path close to the bifurcation (point 2) and further along at $0.6P_{cl}$

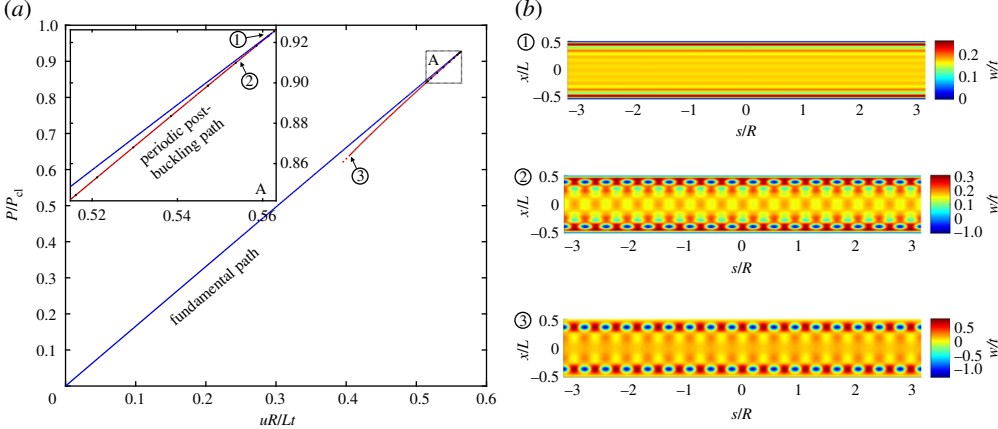

**Figure 2.** (*a*) Pre-buckling (fundamental) and initial periodic post-buckling equilibrium paths of normalized reaction load ($P/P_{cl}$) versus normalized displacement ($uR/Lt$). (*b*) The radial (out-of-plane) displacement $w/t$ over the domain of the cylinder (normalized axial position $x/L$ versus circumferential position $s/R$) for different points 1–3 in (*a*). (Online version in colour.)

(point 3) are also shown in figure 2*b*. The initial post-buckling behaviour (point 2) is governed by the addition of a periodic deformation mode (corresponding to the critical eigenvector). Due to the boundary layer in the initial pre-buckling state, a bias towards the boundaries is observed in the mode shape of point 2. As the post-buckling path is followed further downwards (towards point 3), the periodic post-buckling behaviour concentrates more and more towards the two loaded edges.

As shown in inset A of figure 2*a*, a number of additional bifurcation points (black dots) exist on this periodic post-buckling path. The additional equilibrium paths emanating from these critical points are not shown here, but all cause the buckling mode shape to localize. Similarly, it is well known that the first critical point on the fundamental path is followed by many additional, closely spaced critical points (not shown here). These critical points too connect to periodic post-buckling curves with secondary bifurcation points that lead to localizations of the buckling mode. Hence, as suggested previously [16,18], the subcritical buckling behaviour of the axially compressed cylinder endows it with a strong proclivity towards localization in the post-buckling regime.

The large number of localizations makes it difficult to chose one mode over the others in studying the evolution of the post-buckling behaviour. In the following, we focus on the mountain-pass solution corresponding to a single dimple located at the mid-length of the cylinder ($x/L = 0$) as Horák *et al.* [4] showed this deformation mode to be the easiest escape route to post-buckling. Due to the rotational invariance of the isotropic cylinder, this localization may indeed develop at any position around the circumference ($s/R$) and in each case the equilibrium manifold traced in terms of $P/P_{cl}$ versus $uR/Lt$ is identical. Hence, while a localization at $s/R = 0$ is studied here, the observed behaviour accounts for all single-dimple localizations in the continuous set $s/R \in [-\pi, \pi]$.

## 4. Formation and growth of localized post-buckling equilibria

The localized, unstable single dimple can be determined using a mountain-pass algorithm [4], by tracking dynamical edge states [23], or by perturbing the pre-buckling state of the cylinder with a lateral probing force [7]. Although the latter approach is not as mathematically rigorous as the former two, it is simple to apply within a finite-element framework. When the pre-buckled cylinder is perturbed by a lateral side force prior to the first critical load, the ensuing load–displacement behaviour describes a general softening-stiffening sigmoidal curve (see fig. 7*a* in §5). The probing force increases until it reaches a maximum and then decreases, crossing the state of zero probe force with one negative eigenvalue in the tangential stiffness matrix. This

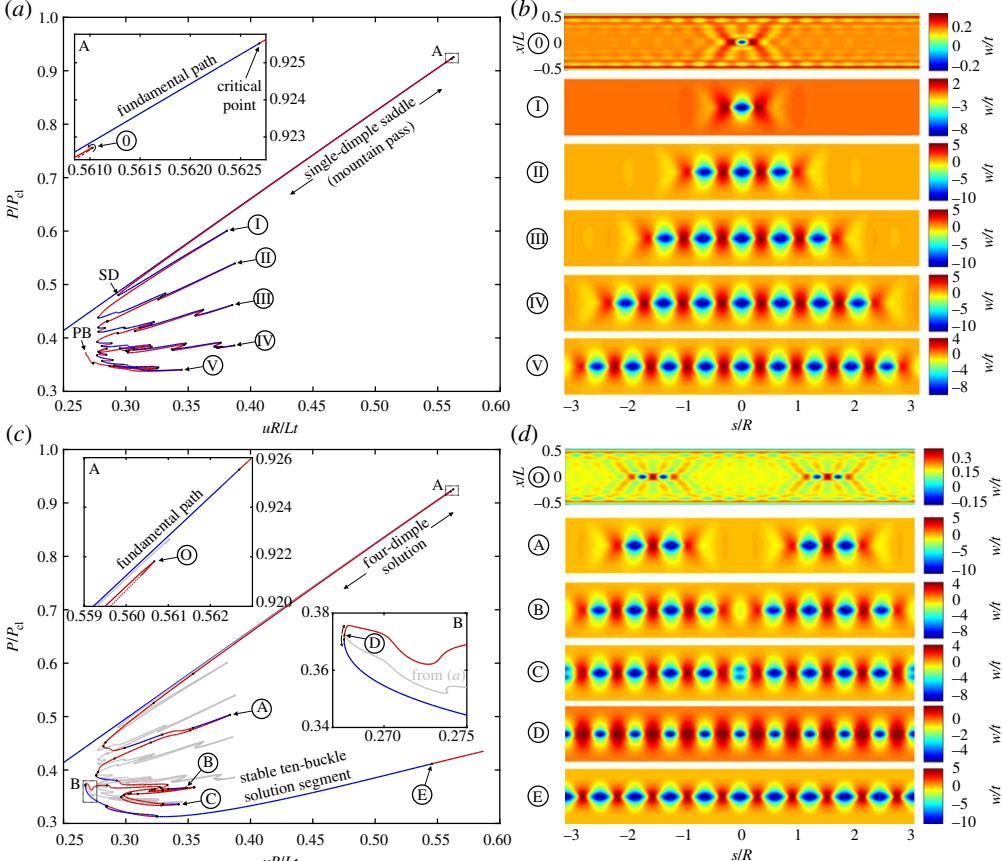

**Figure 3.** (*a*) Equilibrium path of a single-dimple post-buckling solution growing sequentially around the cylinder circumference through a series of destabilizations and re-stabilizations known as homoclinic snaking (or cellular buckling). (*b*) The radial deformation mode shapes over the domain of the cylinder for different points 0–V in (*a*). (*c*) Equilibrium path of a four-dimple post-buckling solution (in red/blue) growing sequentially via snaking. The single-dimple snaking solution from (*a*), shown in grey, connects to this red/blue path at a pitchfork bifurcation (see point D in inset B). (*d*) The radial deformation mode shapes over the domain of the cylinder for different points 0–E in (*c*). (Online version in colour.)

zero-probe-force equilibrium takes the form of a single dimple. Using this probing method we find two localized states: one by probing radially inwards and the other by probing radially outwards. The equilibria found in this way correspond to one- and two-dimple solutions (radially inward dimples), respectively, as shown in figures 3*a* and 4*a*. Once determined, the evolution of these unstable dimple equilibria is evaluated by path-following with respect to the primary parameter: the applied compressive end-shortening.

## (a) From fully localized to axially localized post-buckling behaviour

Figure 3*a* shows the equilibrium path ($uR/Lt$ versus $P/P_{cl}$) of the single dimple and its position relative to the fundamental path. The stable fundamental path runs diagonally in blue with the unstable single-dimple path running almost coincidentally next to it. As shown in inset A, the single-dimple solution does not branch from a bifurcation point on the fundamental path but belongs to a broken-away path delimited by a turning point. This limit point is denoted by point 0 in figure 3*a*. The corresponding radial deformation mode ($w/t$) at this limit point is shown in figure 3*b*. The deformation mode clearly shows a localized dimple in the centre of the domain with some contributions from the periodic buckling mode at the edges (see point 2 in figure 2*b*).

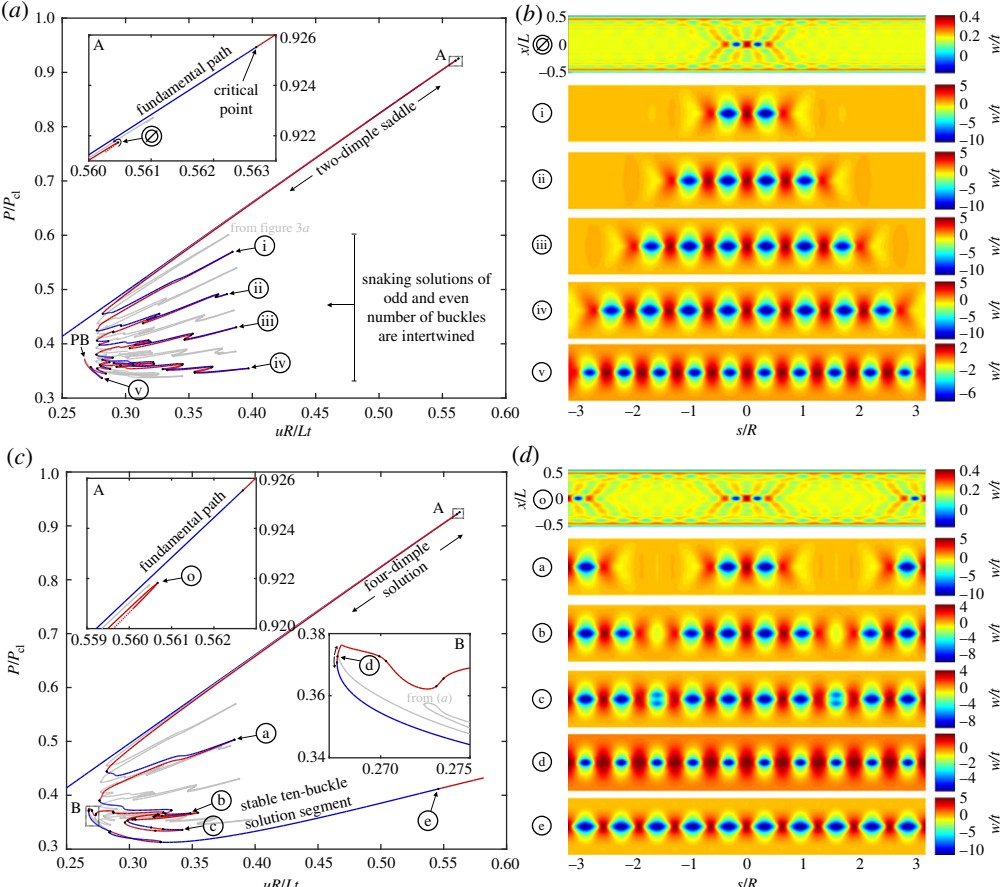

**Figure 4.** (*a*) Equilibrium path of a two-dimple post-buckling solution (in red/blue) growing sequentially around the cylinder circumference through a series of destabilizations and re-stabilizations known as homoclinic snaking. The single-dimple path from figure 3*a* is shown in grey. (*b*) The radial deformation mode shapes over the domain of the cylinder for different points ⌀–v in (*a*). (*c*) Equilibrium path of a four-dimple post-buckling solution (in red/blue) growing sequentially via snaking. The two-dimple snaking solution from (*a*), shown in grey, connects to this red/blue path at a pitchfork bifurcation (see point d in inset B). (*d*) The radial deformation mode shapes over the domain of the cylinder for different points o−e in (*c*). (Online version in colour.)

Path-following one way around the limit point (in the direction of the curved arrow in inset A of figure 3*a*) leads to the snaking sequence previously described by Kreilos & Schneider [23] and which is discussed further below. Path-following in the other direction (in the opposite direction of the curved arrow) leads to a series of additional localizations that are discussed in §5.

Homoclinic snaking describes the sequential destabilization and restabilization of a solution at a series of limit points. In the present case, snaking causes the progressive growth of the single-dimple deformation mode. Starting from limit point 0 (and following the direction indicated by the curved arrow in inset A of figure 3*a*), the single dimple becomes more pronounced with decreasing end-shortening and stabilizes at a second limit point ($uR/Lt = 0.294$, $P/P_{cl} = 0.480$) marked as point SD (single dimple). The value of $P/P_{cl} = 0.480$ exactly matches the numerical result reported by Kreilos & Schneider [23]. This limit point corresponds to the smallest end-shortening for which the single-dimple mountain-pass solution determined by Horák *et al.* [4] exists. The significance of this limit point for design is discussed in §7.

Path-following further, the equilibrium path destabilizes and restabilizes sequentially, adding additional buckles to the left and right of the single dimple. This results in a 'feathered'

equilibrium manifold with six protruding dominant 'fingers'. Each of these fingers corresponds to a unique mode shape of increasing wavenumber. As we proceed along the snaking path, the single dimple grows in the sequence of 1, 3, 5, 7 and 9 waves, until an entire ring around the cylinder is filled with 10 waves at point PB (Pitchfork bifurcation) in figure 3a. The mode shapes corresponding to limit points I–V in figure 3a are shown in figure 3b and clearly show the series of increasing odd buckles (1, 3, 5, 7 and 9) around the cylinder circumference. Interestingly, smaller snaking phenomena also occur on the global snaking 'fingers', with the number of local turning points increasing with the number of buckles on each global snaking finger. These snaking phenomena were not observed in previous work based on Donnell-von Kármán equations [23], and this might be explained by the different kinematics employed by the former moderate rotation and the current large rotation, total Lagrangian approach. The secondary snaking phenomena correspond to the initiation and subsequent decay of additional buckles, but as the corresponding deformations are at least an order of magnitude smaller than the existing buckles, they are unlikely to represent a significant aspect of the pattern formation.

Ultimately, the snaking series ends at a pitchfork bifurcation (point PB) restoring a symmetry group and describing an axially localized, yet axisymmetric buckling pattern with 10 waves around the circumference of the cylinder. Kreilos & Schneider [23] reported a defect in their 10-buckle waveform leading to a weakening of the outer two buckles. As a result, a connection between the snaking path of odd buckles and the circumferentially periodic state with 10 waves could not be determined. On the contrary, we show that a connection between the snaking path of odd buckles and the circumferentially periodic state of 10 buckles does indeed exist via a critical point. As mentioned above, this connection occurs at a pitchfork bifurcation where the snaking solution of odd buckles collides with an equilibrium path that preserves one additional symmetry group. This connection is shown in detail in figure 3c.

In figure 3c, the snaking equilibrium path of odd buckles (figure 3a) is shown in light grey for reference. The additional red/blue equilibrium segment shown in figure 3c, which branches off point PB in figure 3a, is once again delimited by a turning point close the first critical point on the fundamental path (see point O in figure 3c). Point O describes two sets of two dimples located at a quarter-circumference to the left and right of the single-dimple solution (figure 3d). With decreasing end-shortening the two sets of two dimples grow in profile and through a series of de- and re-stabilizations grow from four to eight and, finally, to 10 buckles. The mode shapes corresponding to various points on the equilibrium path of figure 3c are shown in figure 3d and clearly depict the progressive growth of the additional buckles. The two equilibrium paths shown (grey and red/blue) connect at a pitchfork bifurcation (point D in inset B of figure 3c, and equally, point PB in figure 3a). The mode shape at bifurcation point D confirms the 10-buckle waveform without defects. Both equilibrium paths connecting at bifurcation point D are unstable, but in the immediate vicinity of this connection, the 10-buckle equilibrium segment regains stability at a limit point. From thereon ($uR/Lt > 0.268$), the 10-buckle waveform is stable until it destabilizes at a pitchfork bifurcation at $uR/Lt = 0.546$ (point E in figure 3c). The mode shapes of points D and E in figure 3d show that the 10-buckle waveform becomes increasingly pronounced over the stable segment, and finally forms a single ring of axially localized diamonds. The minimum load on the stable segment is $P/P_{cl} = 0.313$, which correlates well with the result reported in [23] ($P/P_{cl} = 0.31$).

In the following, we uncover the existence of a path of even-numbered buckles complementary to the snaking sequence of odd-numbered buckles shown in figure 3a. This complementary solution is obtained by probing outwards, which leads to a localization with two inward dimples (separated by an outwards crest), and an ensuing snaking sequence of even-numbered buckles. Snaking and progressive growth of both even and odd number of localized waves occurs in many systems featuring spatial localization [37]. These two snaking paths are often intertwined and connected by rungs to form a snakes-and-ladders structure. Due to the imposed symmetry conditions in the current quarter-model, connections between the equilibrium paths of odd- and even-numbered buckles can not be shown.

Figure 4*a* shows the equilibrium path of the two-dimple solution (even number of buckles) superimposed on the fundamental path (running diagonally in blue), and, for reference, the snaking solution of odd buckles (from figure 3*a*) in grey. Like the equilibrium manifold in figure 3*a*, this new two-dimple solution is broken away from the fundamental path and is delimited by a turning point close to the first critical point on the fundamental path (see inset A of figure 4*a*). This turning point is denoted as ∅ in figure 4*a* and the corresponding deformation mode is shown in figure 4*b*. The deformation mode clearly shows two localized dimples in the centre of the domain. Most importantly, the equilibrium path of even buckles (red/blue curve) also exhibits snaking and is entangled with the snaking solution of odd buckles (grey curve). The different mode shapes corresponding to limit points i–v in figure 4*a* are plotted in figure 4*b* and show the series of increasing even buckles (2, 4, 6, 8 and 10) growing around the cylinder circumference. Without the imposed symmetry conditions, the odd-buckle path (grey) and even-buckle path (red/blue) would be connected with symmetry-breaking 'ladders'. The two snaking paths are constrained within a corridor of applied end-shortening known as the pinning or snaking region [37].

The snaking solution of even buckles also ends at a pitchfork bifurcation (point PB in figure 4*a*) where it connects to another segment of the equilibrium path that restores one additional symmetry group. This connecting equilibrium path is shown in figure 4*c* with the even-buckle path from figure 4*a* superimposed in grey colour. The red/blue path in figure 4*c* reflects the behaviour previously shown in figure 3*c* with two sets of two dimples growing sequentially from a limit point o (see inset A in figure 4*c*) into four, eight and, finally, 10 buckles. The mode shapes corresponding to various points on the equilibrium path of figure 4*c* are shown in figure 4*d*. The two segments (red/blue and grey) of the equilibrium path connect at a pitchfork bifurcation (point d in inset B of figure 4*c* or point PB in figure 4*a*). The mode shape of point d in figure 4*d* confirms the expected 10-buckle waveform. In conclusion, both the odd- and even-buckle curves (figures 3*a* and 4*a*, respectively) lead to an axially localized post-buckling state of a single ring of 10 diamonds by connecting to other curves (figures 3*c* and 4*c*) that preserve an additional symmetry group.

Due to the rotational symmetry of the isotropic cylinder, the equilibrium curves of the two four-dimple solutions shown in figures 3*c* and 4*c* overlap; the two curves represent four dimples, rotated through 90 degrees, growing into one ring of 10 buckles. Because symmetry conditions are applied to model a quarter-cylinder, the equilibrium curves of the two growing four-dimple solutions (figures 3*c* and 4*c*) suggest different stability (red and blue segments). This only occurs because some symmetry-breaking bifurcations that are possible in figure 3*c* are prevented by symmetry conditions in figure 4*c*. An analysis of the full cylinder would show the equilibrium curves in figures 3*c* and 4*c* to have the same stability characteristics. Extending the analysis to the full cylinder will be a topic of future work.

Finally, we note that Yamaki's experiments on the present cylinder [28] showed initial buckling into an eleven-form buckling mode that then transitioned to 10 and later to nine buckles as end-shortening was increased. It is possible that the buckling mode with eleven waves exists as a separate equilibrium path on the response diagram, but our analyses, and indeed the analyses by Kreilos & Schneider [23], suggest that it is the 10-form mode that lies on one continuous path starting from the single dimple. In the dynamic domain of an actual experiment, the cylinder could, nevertheless, transition and stabilize into this alternative mode of eleven buckles once the first instability arises. Another possibility is that the imposed symmetry boundary conditions prevent bifurcations that lead to a buckling mode with 11-fold periodicity. This possibility provides further motivation to extend the analysis to the full cylinder.

## (b) From axially localized and circumferentially periodic waveform to the Yoshimura pattern

The single ring of 10 diamond-shaped buckles destabilizes at a symmetry-breaking pitchfork bifurcation (point E in figure 3*c* and point e in figure 4*c*). The work by Hunt *et al.* [21] and Kreilos &

**13**

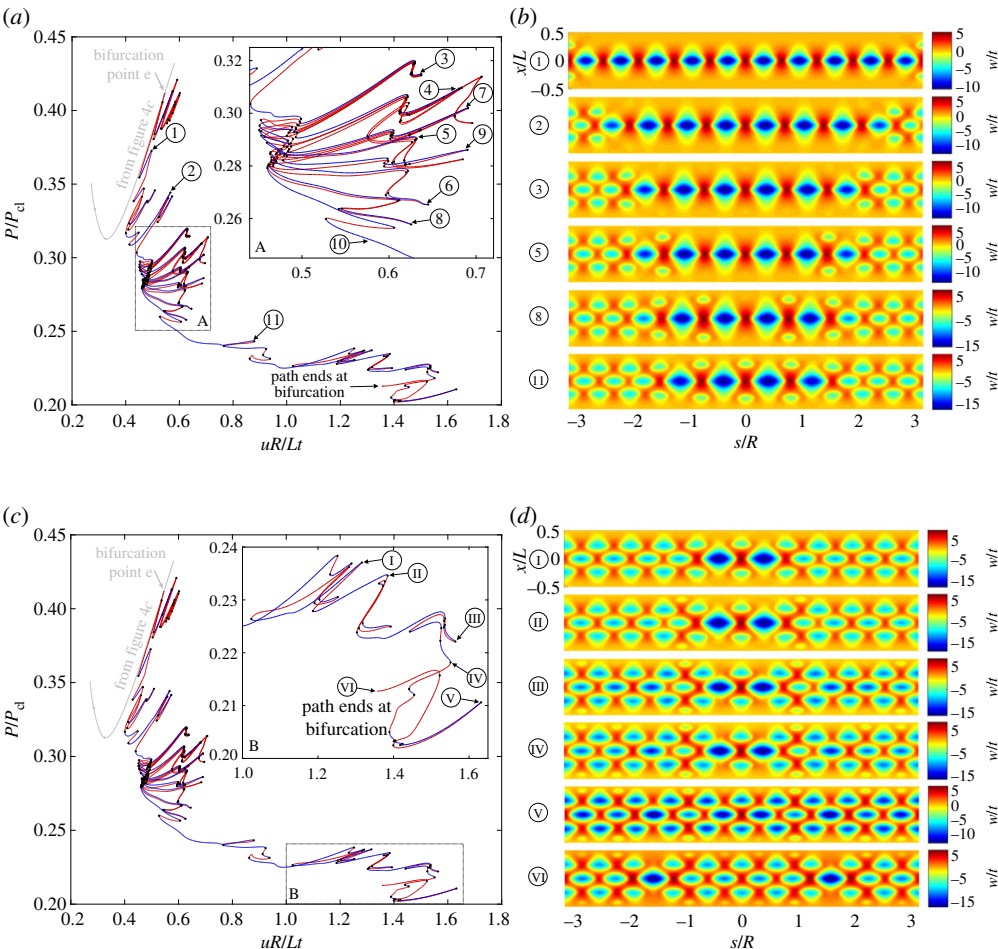

**Figure 5.** (*a,c*) Equilibrium path (in red/blue) branching from a pitchfork bifurcation on the equilibrium path of the single ring of 10 diamonds (in grey). On this red/blue path additional rings of buckles grow sequentially around the cylinder circumference by starting from a single dimple. The two plots (*a,c*) differ in the regions highlighted in insets A and B. (*b*) The radial deformation mode shapes over the domain of the cylinder for different points 1–3, 5, 8 and 11 in (*a*). (*d*) The radial deformation mode shapes over the domain of the cylinder for different points I–VI in (*c*). (Online version in colour.)

Schneider [23] suggests that a second snaking sequence in the axial direction of the cylinder commences once the single ring of diamonds destabilizes. In these works, additional rings of buckles are added in their entirety with each traversal of a limit point. This sequence results in the growth of one ring of diamond-shaped buckles into the full diamond pattern highlighted by Yoshimura [38]. In both [21,23], this second snaking sequence is obtained by imposing a 10-fold circumferential symmetry in the model, such that each additional ring of buckles can only arise as one. With the quarter-model studied here (twofold symmetry), this snaking mechanism with 10-fold circumferential symmetry could not be confirmed. Instead, the previously studied circumferential pattern formation that underlies the first ring of 10 buckles, is repeated for all additional rows. Hence, each additional ring does not appear at once, but grows circumferentially from an initial seed of two dimples, one above and one below the existing ring(s) of buckles. In this view, circumferential snaking is the governing mechanism behind the pattern formation along the axis of the cylinder—first forming one ring and then leading to further additions.

This mechanism is summarized graphically in figure 5. Figure 5*a,c* shows the same equilibrium path (in red/blue) branching from the symmetry-breaking bifurcation (point e in figure 4*c*), with the last segment of the 10-diamond equilibrium segment shown in grey. The difference between

figures 5*a*, *c* is that they highlight different portions of the red/blue equilibrium curve in insets A and B, respectively. Some of the deformation mode shapes corresponding to points 1–11 in figure 5*a* and points I–VI in figure 5*c* are shown in figure 5*b*,*d*, respectively. For completeness, additional mode shapes are shown in appendix A. It is apparent that the equilibrium manifold is complex with many entangled segments that loop back on themselves. We focus here on the patterns formed in insets A and B as these represent most of the interesting behaviour. For general guidance, the points 1–11 and I–VI are numbered in increasing order as they appear when path-following the equilibrium curve from the top left to the bottom right of figure 5.

As shown in figure 5*a*,*b*, two new rings of buckles are initiated by dimple localizations above and below the fully formed ring of 10 diamond-shaped buckles. Through a snaking sequence, these new dimples multiply circumferentially until they fill the whole circumference (see point 11). In the beginning, these dimples squeeze the original ring of 10 diamonds into smaller shapes. The snaking sequence in inset A ends in a long stable plateau governed by the mode shape corresponding to point 11 in figure 5*b*, which shows that four of the original 10 diamonds persist in their original size once the new rings are completely formed. Inset B in figure 5*c* and the corresponding mode shapes in figure 5*d* then show how the fourth and fifth rings of buckles form. Point I shows the fully formed second and third rings with only two of the diamonds in the central ring of buckles maintaining their original size. The fourth and fifth rings are then initiated by additional localizations (point II), which grow sequentially to almost completely fill the circumference (point V). The unstable equilibrium state at point VI is a pitchfork bifurcation where the red/blue equilibrium path depicted in figure 5 collides with another equilibrium path that restores one additional symmetry group. At this point the post-buckled state is almost entirely symmetric apart from two small defects (see point VI in figure 5*d*).

This additional equilibrium path with one additional symmetry group is depicted by the red/blue curve in figure 6, with the previous equilibrium path of figure 5 shown in grey. The bifurcation point where the two paths (grey and red/blue) intersect is clearly marked as point VI in inset A of figure 6*a*. The red/blue equilibrium path in figure 6*a* is delimited by a turning point in the vicinity of bifurcation point e in the top left corner of the figure. The second and third, and fourth and fifth rings are again formed by snaking, but in this case, the pattern formation occurs symmetrically rather than from one side (see the deformation modes of points i–iv in figure 6*b*). This post-buckled path reaches a long plateau of stability starting from ($uR/Lt = 1.41$, $P/P_{cl} = 0.201$) and the deformation mode shape along this stable path resembles the Yoshimura pattern (see point v in figure 6*b*).

In summary, the present analysis shows that growth of localized waveforms in the axially compressed cylinder is driven by a repeating sequence of circumferential snaking. Once a single-dimple forms, it can grow circumferentially within a bounded range of end-shortening and decreasing force, until one ring of diamond-shaped buckles is complete. Upon further loading, this single ring of diamonds destabilizes at a symmetry-breaking bifurcation that initiates the growth of additional diamond rings above and below the previously formed ring of buckles. Rather than appearing at once, these additional rings are also seeded by single dimples that then grow around the cylinder. The post-buckling patterns with greater number of axial rings occur for higher levels of end-shortening.

Due to the nature of snaking, the pattern formation is governed by multiple stable and unstable paths. In the naturally dynamic setting of an experiment, the quasi-static progression of growing localizations is rarely observed. Rather, individual rings of buckles may pop into existence partially, or in their entirety, depending on which equilibrium branch of the snaking manifold the system stabilizes on. As shown in figures 4*c* and 5*a*: (i) the first critical load on the pre-buckling path; (ii) the stabilized single ring of 10 diamonds; and (iii) the snaking sequence to higher number of rings, all occur around $uR/Lt \approx 0.55$. Precisely manufactured cylinders that buckle close to the idealized critical load could therefore snap straight into a partially formed single ring of diamonds; surpass the first snaking sequence and stabilize in one ring of buckles; or indeed settle on one of the partially formed states with multiple rings. During the dynamic

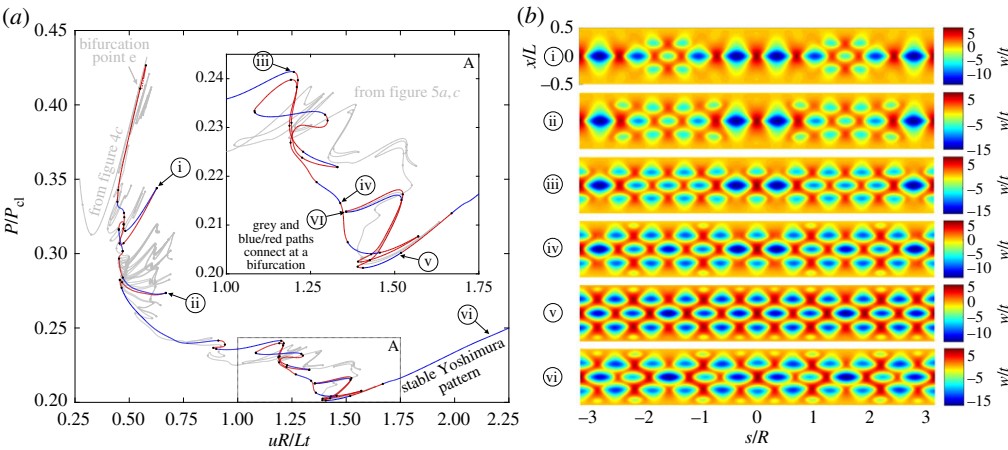

**Figure 6.** (a) Equilibrium path describing the formation of additional rings of buckles from four localizations, i.e. the pattern formation occurs symmetrically around the circumference rather than from one side (as is the case in figure 5). (b) The radial deformation mode shapes over the domain of the cylinder for different points i–vi in (a). (Online version in colour.)

snapping sequence, many of the intermediate states of partially formed rings may be visible if the dynamic snap passes through, but does not stabilize on, these equilibria.

As the buckling event is governed by time-frames in the 1/1000 of a second [12], special equipment is generally required to observe the pattern formation. High-speed photography [12] and dynamic analyses [39] indicate that a single buckle generally initiates buckling, which then spreads both circumferentially and axially. In fact, high-speed photography experiments by Tennyson [13] suggest that buckling initiation is a 'very localized phenomenon' with the buckling waveform then 'propagat[ing] circumferentially in both directions'. Often, the circumferential growth of the buckling pattern is hidden in the dynamics of the buckling event. For shorter cylinders, however, the pattern formation has been statically observed. For a short cylinder with Batdorf parameter $Z = 20$, Yamaki remarked that 'a local buckle first appeared and then propagated circumferentially with increasing edge shortening' [28, p. 226]. In conclusion, the predominance of circumferential snaking described herein is qualitatively reflected in various experimental studies.

## 5. Stability landscape and energy barriers

As mentioned in §1, the unstable equilibrium with the lowest energy barrier straddling pre-buckling and post-buckling valleys is known as a mountain-pass point. Horák *et al.* [4] showed that the mountain-pass point of the axially compressed cylinder describes a single dimple. To study the stability landscape surrounding the pre-buckling equilibrium, recent experimental studies have probed axially compressed cylinders with a lateral indenter [22]. Here, we perform such an experiment numerically and inquire further with regards to some of the distinctive features of the stability landscape. The analysis involves two fundamental parameters: the applied end-shortening ($u$) and a lateral probing force ($F$) acting at right angles to the cylinder mid-surface, half-way along the cylinder length. In the following analysis, the cylinder is loaded to a specific level of end-shortening, at which point, a lateral side force is applied to quantify the resilience of the pre-buckled state to perturbations.

Probing the cylinder from the side leads to a general nonlinear softening/stiffening relationship between the probe force and the ensuing dimple displacement. Such equilibrium curves for different levels of axial compression are plotted in figure 7a in terms of normalized probing force $FR/Et^3$ versus normalized radial displacement $\Delta w/t$ at the probe point. The $\Delta$-symbol denotes that the radial displacement is taken relative to the radial Poisson's dilation

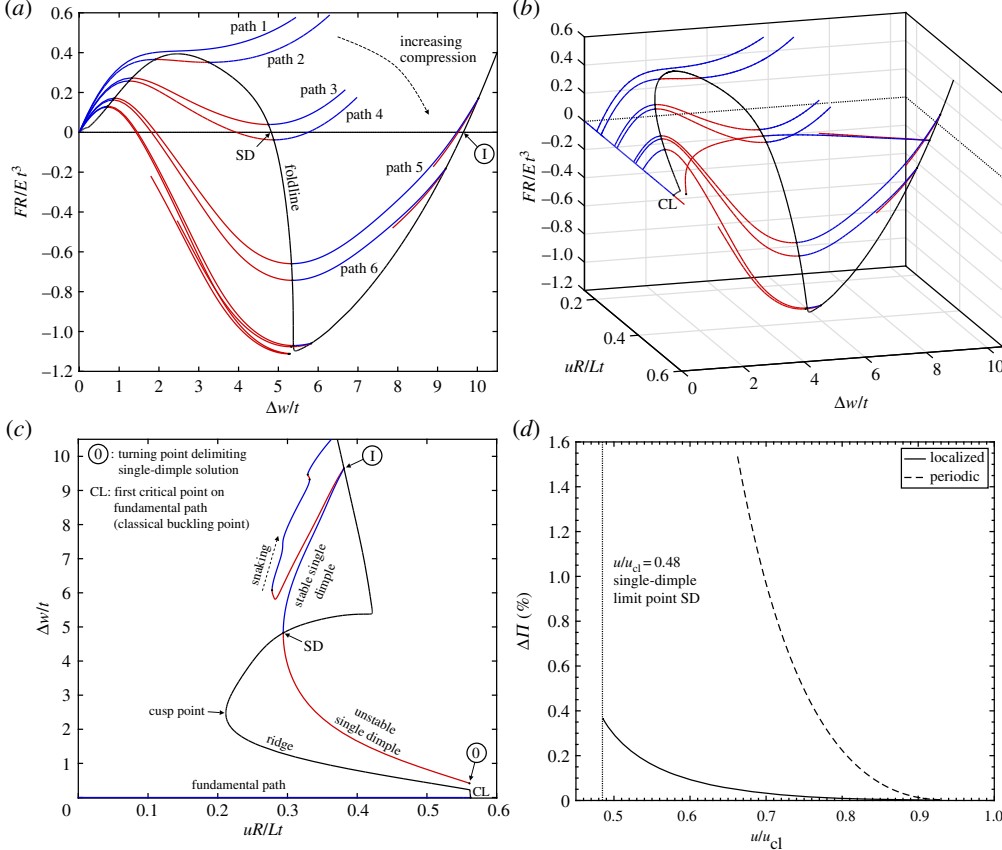

**Figure 7.** Stability landscape of an axially compressed cylinder with a probing side force. (*a*) Normalized probe force ($FR/Et^3$) versus normalized probe displacement ($\Delta w/t$) for different values of normalized axial compression ($uR/Lt$). (*b*) The stability landscape in terms of axial compression versus probe force versus probe point displacement. (*c*) Normalized axial compression versus normalized dimple amplitude of the fundamental and single-dimple equilibrium paths. (*d*) The energy barrier ($\Delta \Pi$) with varying axial compression ($u/u_{cl}$) between the pre-buckling state and: (i) the unstable single dimple and (ii) the unstable periodic mode (figure 2). (Online version in colour.)

occurring in the pre-buckling state; i.e. $\Delta w = 0$ for all $u$ along the fundamental path. For low levels of axial compression, the cylinder resists the lateral force with nonlinear, but always positive stiffness (path 1 in figure 7*a*). For greater levels of end-shortening, the equilibrium manifold traces sigmoidal curves, but as the dimple forms, the lateral stiffness of the cylinder reduces, until limit points are traversed leading to a region of negative stiffness (see paths 2–3). Upon reaching the maximum load in a force-controlled experiment, the radial perturbation would thus cause a small 'pop' as the dimple increases in size with constant probe force. For further increasing end-shortening, the probing force along the unstable portion of the curve reduces sufficiently to pass through the zero load axis (e.g. $F = 0$ on path 4). At the point where $F = 0$, an unstable equilibrium in the shape of a single dimple has been found, and because the tangential stiffness matrix at this state only has one negative eigenvalue, the point is confirmed to be a saddle on the energy landscape. Within a specific range of axial compression, the curve re-emerges through the $F = 0$-axis in the form of a stable dimple (see paths 4–5). Also shown in figure 7*a* is a black foldline that connects the maximum and minimum limit points for different levels of end-shortening, and therefore represents a ridge that separates the domain into stable and unstable regions.

When expanding figure 7*a* in $uR/Lt$ for a three-dimensional view, an interesting stability landscape emerges, which is shown in figure 7*b*. The landscape matches qualitatively the experimental landscape produced by Virot *et al.* [22]. The area between the stable pre-buckled

and unstable single-dimple solutions under the $F$ versus $\Delta w$ curve represents the energy barrier that needs to be overcome to escape the pre-buckling energy well. The size of this energy barrier can be intuited by the maximum force on each $F$ versus $\Delta w$ curve of figure 7a. The foldline in figure 7b connecting these maximum points slopes down towards the first critical point on the fundamental path (point CL). Indeed, the foldline intersects the first critical point on the fundamental path confirming that the resilience of the pre-buckling state to small perturbations (i.e. stability) vanishes at the first critical point.

Figure 7c shows a cross section of figure 7b for $F = 0$. This highlights the path of the single-dimple solution alongside the fundamental equilibrium path, which was previously shown in figure 3a with $P/P_{cl}$ on the vertical axis. The points where $F = 0$ on the foldline (point SD and point I in figure 7a,c) correspond to the limit points that bound the stable segment of the single-dimple equilibrium curve. By tapping axially loaded cylinders with a finger, Eßlinger & Geier [12] were able to determine the stabilized single-dimple post-buckling state. For low levels of compression ($\approx 50\% u_{cl}$) the single dimple popped into existence and then remained stable with increasing end-shortening until a snap to a higher-order mode occurred. For greater levels of end-shortening ($\approx 70\% u_{cl}$) the cylinder collapsed into a fully formed buckling pattern once tapped. Although these experiments were performed on different cylinders, the qualitative observations correlate well with the existence of a stable region of the single dimple between $\approx 50\% - 70\%$ of the classical load, as shown in figure 7c.

Another interesting feature is that the cusp point (hysteresis point) on the foldline takes a value of ($uR/Lt = 0.212$, $P/P_{cl} = 0.350$), which correlates very closely with the lower-bound SP-8007 knockdown factor [9] of $P/P_{cl} = 0.354$ for the dimensions of the present cylinder ($R/t = 405$). This correlation could be explained as follows. The cusp represents the end-shortening for which the perfect cylinder loses the ability to resist a probing disturbance with exclusively positive stiffness. Beyond this point, a lateral disturbance can induce dynamic snaps and the induced kinetic energy may, essentially, provide enough impetus to transition the cylinder into a lower-energy state, thereby triggering the premature collapse captured by the SP-8007 recommendations. At levels of compression coinciding with the cusp ($P/P_{cl} = 0.350$), a lateral force of $F = 0.330$ N is required to induce dynamic snaps, and this force reduces further for increased end-shortening beyond the cusp. At the cusp, the lateral perturbation required to reach the snapping point ($F = 0.330$ N) is thus three orders of magnitude smaller than the applied axial load ($P = 417$ N). Indeed, in statistical mechanics and stochastic systems, where instabilities can be triggered prematurely by statistical fluctuations or disturbances, the cusp signifies the earliest point that a system can transition from one state to another (e.g. [40, p. 53]). Whether stochastic fluctuations or disturbances are sufficiently pronounced in experimental buckling tests to give any credence to this interpretation remains to be seen.

To further quantify the resilience of the pre-buckling equilibrium to external disturbances, we study the energy barrier surrounding the pre-buckling state. In particular, we focus on the single-dimple saddle, which Horák et al. [4] show to be the smallest perturbation that can push the cylinder out of its pre-buckling energy well. The energy barrier—expressed as a percentage difference between the total potential energy of the pre-buckling equilibrium and the unstable single-dimple equilibrium—is plotted in figure 7d as a function of normalized end-shortening ($u/u_{cl}$). For comparison, the percentage energy barrier provided by the unstable periodic post-buckling mode of figure 2 is also shown. While both energy barriers fall off rapidly with increasing compression, the energy barrier of the single-dimple state is typically an order of magnitude smaller than the energy barrier corresponding to the periodic mode. This is not surprising as the localized dimple deforms a considerably smaller portion of the cylinder wall than the periodic mode. Second, over the range of end-shortening for which the single-dimple solution exists ($u/u_{cl} > 0.48$), the energy barrier provided by the single-dimple saddle is less than 0.4% of the energy stored in the pre-buckling state. How easily this energy barrier is eroded by initial imperfections will be the topic of future work, but it is clear that the mountain-pass state does not provide much resilience to disturbances and imperfections once it exists beyond 48% of the classical buckling load.

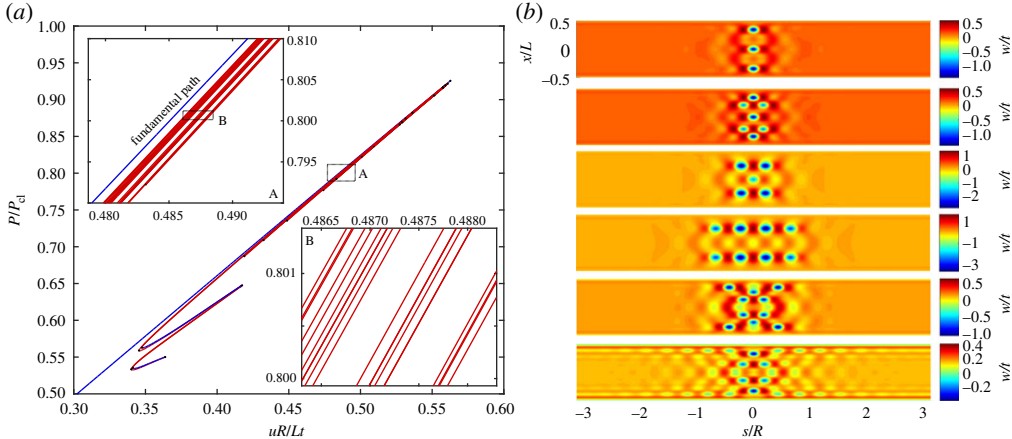

**Figure 8.** Path-following the single-dimple equilibrium path in the opposite direction of the snaking behaviour described in figure 3a gives rise to (a) many additional unstable localizations as the equilibrium path winds up and down parallel to the pre-buckling curve. The deformation shapes corresponding to some of these edge states are shown in (b) and figure 11b, and all represent localized post-buckling modes. (Online version in colour.)

In fact, many additional localized post-buckling modes exist in the vicinity of the pre-buckling energy well. As shown in figures 3a and 7c, the single-dimple equilibrium path originates at a limit point broken away from the pre-buckling path. Continuing this path in one direction leads to the snaking sequence discussed previously in §4. Path-following in the opposite direction reveals many additional unstable localizations in the post-buckling regime. As shown in figure 8a the equilibrium path winds up and down continuously traversing a whole plethora of additional limit points whose eigenvectors are associated with different localized shapes. Only a small subset of the additional localizations found while path-following are shown in figure 8b and further in appendix A (figure 11b). The equilibrium paths in figure 8a show a dense set of curves in the immediate vicinity of the pre-buckling path. Zooming-in on a specific region via insets A and B reveals a fractal-like nature of these equilibrium paths. Due to practical time constraints, the path-following procedure was terminated but additional unstable equilibrium segments and localizations could have been determined by path-following further. Equally, limit points o, O, ∅ shown in figures 3 and 4 can be traversed in the opposite direction to produced more localizations similar to those of figure 8a. These solutions are not shown here for brevity.

Due to the localized nature of the equilibrium paths plotted in figure 8a, these unstable states all represent relatively small energy barriers with respect to the pre-buckling energy well. What is more, each of these unstable equilibria does not correspond to a unique deformation state because the localized nature and rotational invariance of the isotropic cylinder mean that the deformations can occur with equal likelihood anywhere around the circumference. As initially suggested by Horák et al. [4], the basin of attraction of the pre-buckling state is seemingly bounded by a large set of edge states that are easily eroded by initial imperfections or traversed by disturbances. As each type of geometric, material and loading imperfection may push the system towards one of these many competing edge states, the situation as described by 'spatial chaos' [17] may be an appropriate descriptor, in that the observed instability behaviour is highly sensitive to the precise initial conditions.

One further implication of this reasoning is that deterministically applied imperfections bias the cylinder towards one particular post-buckling state, and should therefore improve the correlation between experiment and prediction. Indeed, recent computational approaches that account for the exact nature of imperfections have achieved excellent correlation with experiments on spherical shells [8].

In the absence of such precise information or engineered imperfections, estimates for premature buckling are needed. In §7, we derive a design load based on the existence of the

single-dimple as a mountain-pass point, i.e. the level of compression for which the *smallest* of perturbations can cause the cylinder to transition from the pre-buckling equilibrium to a lower-energy post-buckling state. First, however, we relate mountain-pass points to two other terms from dynamical systems theory that are increasingly being used in contemporary research on cylinder buckling: metastability and the Maxwell energy criterion.

## 6. The Maxwell energy criterion, metastability and mountain-pass points

In classical structural mechanics, the stability of an equilibrium is governed entirely by the Hessian of the total potential energy: a local minimum with respect to all degrees of freedom (energy well) denotes a stable equilibrium, whereas a local maximum in at least 1 d.f. denotes an unstable equilibrium. The stability criterion thus rests in the local nature of the energy landscape. Along an equilibrium path, transitions from one state to another are only possible when a critical point is traversed—causing either a dynamic snap (e.g. at a limit point) or a smooth transition (e.g. at a supercritical pitchfork bifurcation). This framework does not address scenarios in which two equilibrium paths are separated by a small energy barrier but never strictly intersect at a bifurcation point. For example, Hunt *et al.* [41] discuss a model of a 'perfect' layered geological structure where a stable equilibrium (energy well) and an unstable equilibrium (energy saddle) approach asymptotically for increasing load but never intersect at a bifurcation. In this case, a classical stability analysis based on an eigenproblem is of little use as the critical load is theoretically infinite. For increasing load, the trivial state does, however, rest in a state of metastability because small disturbances or geometric misalignments can trigger an escape from the stable equilibrium well over the adjacent unstable saddle.

To address this problem of converging, yet non-intersecting equilibrium paths in the axially compressed cylinder, Hunt & Lucena Neto [42] adopt the Maxwell energy criterion. The Maxwell energy criterion is typically used in thermodynamics, statistical mechanics and stochastic systems to address instabilities triggered by fluctuations, disturbances or excitations (e.g. [40, p. 53]). In this setting, stability is not only governed by local energy wells but also by the global minimum of the total potential energy and the size of energy barriers between these minima. Thus, stability does not only rest in the local topology of the energy landscape but also in its global features. The Maxwell criterion is met when the energy well of a stable post-buckled state ($\Pi_1$) first falls below the energy well of a fundamental state ($\Pi_0$), i.e. when $\Pi_0 = \Pi_1$. For the axially compressed cylinder, the Maxwell load (displacement) is the force (end-shortening) for which the pre-buckled state and any of the restabilized post-buckling states have equal total potential energy.

The Maxwell criterion of equal energy between pre-buckled and post-buckled states also implies the existence of a mountain-pass point between these states. The mountain-pass theorem guarantees the existence of a saddle point in a nonlinear functional given certain pre-requisites. Apart from a suitable functional ($\Pi$), the key requirements for a mountain-pass point are: (1) the existence of a local minimum $S_0$, and (2) the existence of a second point $S_1$ (not necessarily a minimum) such that $\Pi(S_0) \geq \Pi(S_1)$. In the case of the cylinder, this translates to the situation where a post-buckled state (stable or unstable) exists with lower levels of total potential energy than the stable pre-buckled state. This situation is first possible when $\Pi(S_0) = \Pi(S_1)$—i.e. when the Maxwell criterion is met.

In summary, there is a direct link between the Maxwell energy criterion, metastability and mountain-pass points; three concepts that have received increasing attention in contemporary perspectives on the classical problem of cylinder buckling. Namely, (i) the existence of a mountain-pass point (for a specific level of end-shortening) implies, by definition, that a post-buckling state of lower energy than the fundamental state exists, such that (ii) the cylinder can, in principle, be perturbed out of its fundamental state and transition to a lower energy state—i.e. the pre-buckling state is defined to be metastable—which guarantees that (iii) the Maxwell energy criterion of equal fundamental and post-buckling energies must be passed.

The work of Horák *et al.* [4] showed that the single dimple is the smallest mountain-pass point for the axially compressed cylinder, i.e. the lowest saddle point straddled between the

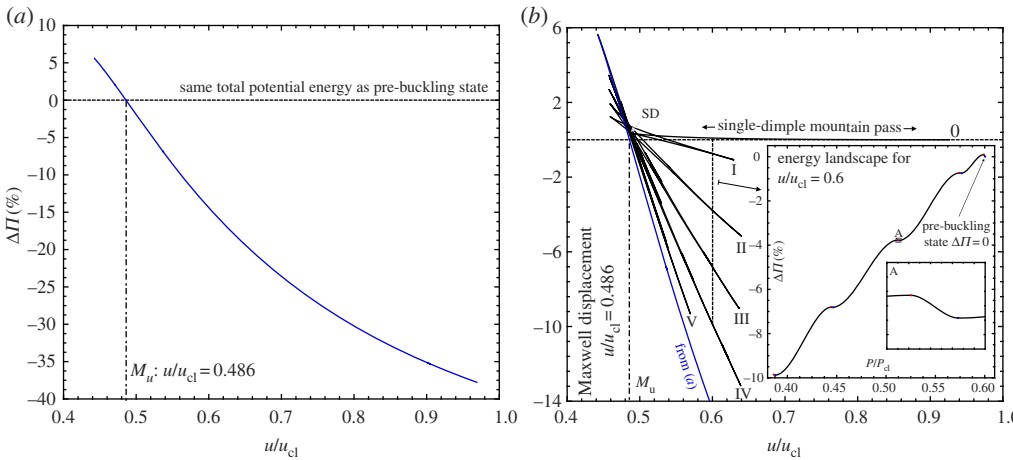

**Figure 9.** Percentage difference in total potential energy ($\Delta\Pi$) with varying end-shortening ($u/u_{\text{cl}}$) between: (*a*) the pre-buckling state and the single ring of 10 circumferential diamonds (figure 3*c*) and (*b*) the pre-buckling state and the snaking equilibrium path of odd buckles (figure 3*a*). The Maxwell displacement, $M_u$, is highlighted in (*a*) and occurs when the energy levels on the pre-buckling and post-buckling paths are first equal. Figure (*b*) also shows the energy landscape at a constant end-shortening of $u/u_{\text{cl}} = 0.6$, demonstrating the rapidly decreasing load-carrying capacity of post-buckled states as the single-dimple energy barrier next to the pre-buckling well is traversed. (Online version in colour.)

valley around the pre-buckled state and many different valleys in the post-buckling regime—even though other mountain-pass points (e.g. the two-dimple localization shown in figure 4*a*) also exist as connections to further post-buckling states.

Figure 3*a* shows that the single-dimple saddle first exists for levels of end-shortening beyond limit point SD. In the following, we compare limit point SD to a Maxwell energy prediction. One difficulty in applying the Maxwell energy criterion is that a whole plethora of stable post-buckling states exist that can be compared with the fundamental path. Of the restabilized post-buckling states that have been obtained in this study (figures 3–6), we choose the axially localized/circumferentially periodic mode of one ring of 10 diamonds (see segment ending in point E in figure 3*c*) because this mode exists for the lowest value of end-shortening and the single-dimple snaking path connects to this state. As described above, many different mountain-pass points and restabilized post-buckling states exist, and therefore it is not guaranteed that a Maxwell energy prediction based on the chosen post-buckling path coincides with limit point SD. However, given that the Maxwell energy criterion can be interpreted as an organizing centre for snaking (see Knobloch [37]) we should expect the Maxwell energy prediction based on the 10-buckle pattern to fall in the close vicinity of limit point SD.

Figure 9*a* shows the total potential energy on the stable segment of the 10-buckle path plotted against normalized end-shortening ($u/u_{\text{cl}}$). The total potential energy is expressed as a percentage difference ($\Delta\Pi$) between the energies of the pre-buckling state and the single ring of 10 buckles. For low levels of compression, the single ring of 10 diamonds has greater levels of total potential energy than the fundamental state (see $\Delta\Pi > 0$ in figure 9*a*). When $\Delta\Pi = 0$, the energy levels of the two modes are equal, such that the Maxwell stability criterion is met. As load is applied in a rigid manner (end-shortening), we find a Maxwell *displacement* of $M_u = u_{\text{M}}/u_{\text{cl}} = 0.486$ ($u_{\text{M}}R/Lt = 0.294$). As expected, this value correlates closely with the limit point SD ($u_{\text{SD}}/u_{\text{cl}} = 0.480$) highlighted in figure 3*a*. Beyond the Maxwell displacement, the total potential energy of the single ring of 10 buckles is always lower than the pre-buckling solution. Beyond $M_u$, end-shortening is therefore accommodated more efficiently by developing a localized region of bending-dominated deformation than by the pre-buckling state dominated by membrane action.

Figure 9*b* plots the percentage difference between the total potential energy of the pre-buckling state and the snaking path of odd buckles (figure 3*a*) against normalized end-shortening. The

equilibrium manifold of this path was previously shown in figure 3*a* and the corresponding points 0–V are repeated in figure 9*b*. As discussed before, the snaking path of odd buckles connects to the 10-buckle mode at a pitchfork bifurcation. There are a number of points on the snaking path of odd buckles where $\Delta\Pi = 0$, but each of the zeros (corresponding to 1, 3, 5, 7 and 9 buckles) occur for displacements ($u/u_{cl}$) greater than the computed Maxwell displacement ($M_u = 0.486$).

The inset in figure 9*b* shows the total potential energy of the stable and unstable equilibria that exist on the snaking path for a level of end-shortening of $u/u_{cl} = 0.6$. The stable and unstable equilibria are denoted by blue and red dots, respectively, and on a plot of energy ($\Delta\Pi$) *versus* normalized reaction force ($P/P_{cl}$), correspond to energy minima and maxima, respectively. The black curve connecting these points is not computed directly, but for the purpose of illustration, interpolated in Matlab using the `pchip` function. The pre-buckling energy well is located in the top right-hand corner of the inset and bounded by a small energy barrier corresponding to the unstable single-dimple saddle. All other equilibria occur for energy levels considerably lower than the pre-buckling state. Hence, if the small energy barrier of the single-dimple saddle is traversed, the system can easily cascade down the energy landscape and restabilize in a post-buckling state of smaller load-carrying capacity ($P/P_{cl}$). In other words, initiating the first dimple is the main obstacle to lower energy states, after which further dimples are readily added without having to increase the total potential energy.

In the following section, we use the point where the single-dimple mountain-pass point first exists, and dynamics to lower energy wells for 'small perturbations' are likely, to derive an estimate for the buckling of imperfect cylinders. This compressive boundary demarcating an increased sensitivity to perturbations was termed 'shock sensitivity' by Thompson & van der Heijden [43]. The reason that we focus on the single-dimple mountain-pass point, and not any other, is that it provides the smallest energy barrier against perturbations. Furthermore, the single-dimple mountain pass describes a sensitivity to the simplest of all stimulating perturbations—a single concentrated imperfection or disturbance. Finally, the single dimple has also arisen in a number of different settings: (i) as the initial buckling mode in experiments [12,13]; (ii) as the 'worst' imperfection in a buckling load minimization over all possible imperfections [44]; and (iii) as a stimulating imperfection [45], i.e. a perturbation that excites the characteristic buckling behaviour observed in experiments.

## 7. Buckling design load based on shock sensitivity

It has been argued [45] that the well-known lower-bound curve of NASA's SP-8007 guideline [9] is overly conservative for shells manufactured using modern manufacturing techniques. We, therefore, attempt to derive a less conservative alternative based on the idea of shock sensitivity and the existence of a mountain-pass point. As described in §6, the existence a mountain-pass point implies that the cylinder can readily be perturbed into a lower-energy post-buckling state, and the single-dimple mountain-pass point derived by Horák *et al.* [4] first occurs at limit point SD (shown in figure 3*a*). Hence, to define the critical boundary of shock sensitivity for different shell geometries, we trace limit point SD with respect to parameters $t$, $R$ and $L$.

Figure 10*a* shows the foldlines depicting the evolution of SD. The parameter $a/a_0$ on the *x*-axis defines the value of each geometric parameter $a = t$, $a = R$ and $a = L$ with respect to the baseline values of $t_0 = 0.247$ mm, $R_0 = 100$ mm and $L_0 = 160.9$ mm. To fit all three curves within the same range of *y*-axis values, the end-shortening at the single-dimple critical point, $u_{sd}$, has been multiplied by a factor of three and 10 for $a = R$ and $a = L$, respectively.

A power law regression of the foldlines suggests that the end-shortening at limit point SD varies approximately as $u_{sd} \overset{\propto}{\sim} t^{1.16}$, $u_{sd} \overset{\propto}{\sim} R^{-0.84}$ and $u_{sd} \overset{\propto}{\sim} L^{0.68}$ (to two decimal places with $R^2 > 0.99$ in all three cases). In combination, this regression suggests that $u_{sd}$ follows a foldline that compares to the classical prediction ($u_{cl} \propto Lt/R$) as follows:

$$\frac{u_{sd}}{u_{cl}} = \frac{P}{P_{cl}} = 1.48\frac{R^{0.16}t^{0.16}}{L^{0.32}} = 1.48\eta, \tag{7.1}$$

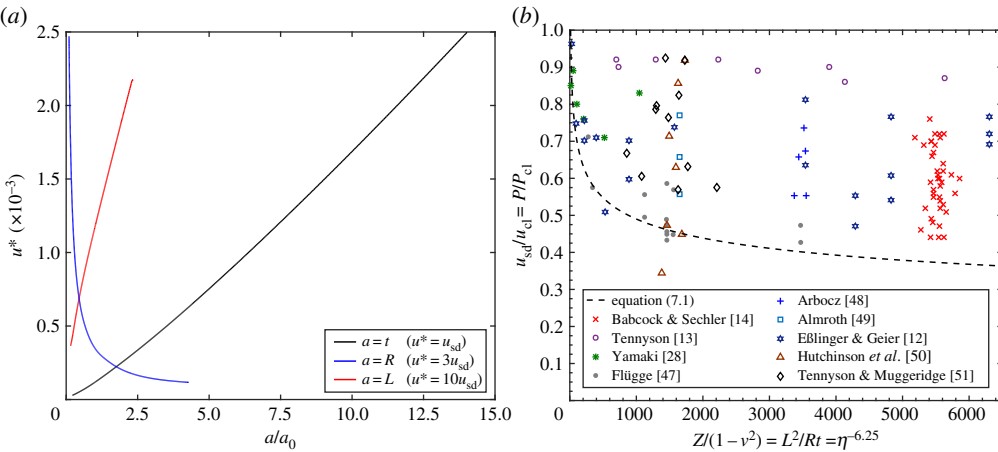

**Figure 10.** (a) Critical boundaries denoting the evolution of the single-dimple limit point SD with respect to geometric parameters of thickness ($a = t$), radius ($a = R$) and length ($a = L$). For $a/a_0 = 1$, we have the baseline parameters $t = 0.247$ mm, $R = 100$ mm and $L = 160.9$ mm. (b) Power law regressions of the curves in (a) lead to the critical curve expressed by equation (7.1) and this is compared against 115 experimental data points from the literature. (Online version in colour.)

where $\eta = R^{0.16}t^{0.16}L^{-0.32}$ is a non-dimensional parameter that describes the onset of shock sensitivity with respect to the classical compression at buckling. Note that due to the linearity of the pre-buckling path, the normalized end-shortening that denotes the onset of shock sensitivity ($u_{sd}/u_{cl}$) is equal to the normalized reaction force $P/P_{cl}$. It is striking that there is an apparent power-law relation between the parameter $\eta$ and the well-known Batdorf geometric parameter $Z$, i.e. $Z/\sqrt{1 - v^2} = L^2/Rt = \eta^{-6.25}$. In fact, Batdorf et al. [46] showed in the 1940s that experimental buckling loads are a function of $Z$ and not only of $t/R$ as classical theory suggests. The critical boundary expressed by equation (7.1) is drawn on a plot of $u_{sd}/u_{cl}$ versus $\eta^{-6.25}$ in figure 10b alongside a set of 115 experimental data points taken from nine different studies in the literature [12–14,28,47–51].

Figure 10b shows that the experimental data points fall on or above the shock sensitivity knockdown curve of equation (7.1) with the exception of one outlier from the study by Hutchinson et al. [50]. Many of the studies listed are efforts from the 1960s and 1970s that explicitly set out to manufacture cylindrical shells with geometric tolerances as 'nearly perfect as possible', e.g. [13,14,28,48]. For example, the manufacturing method chosen by Babcock & Sechler controlled the radial dimension to the order of half the wall thickness and constrained thickness variations to less than 3%. For these cases, the experimental buckling load can be seen to fall well above the critical boundary of shock sensitivity. In the study by Babcock & Sechler [14], the 'nearly perfect' cylinders buckled between 72 and 76% of the classical load, and the experiments by Yamaki [28] and Tennyson [13] even show experimental buckling loads within 90% of the classical load.

The set of data points also include studies with engineered imperfections, e.g. [14,50,51]. Babcock & Sechler [14] imposed axisymmetric imperfections in the form of a half sine wave (radially inwards and outwards) along the cylinder length, and Tennyson & Muggeridge [51] considered higher-order periodic axisymmetric imperfections. In both cases, all data points fall above the shock sensitivity curve. Hutchinson et al. [50] imposed local axisymmetric imperfections (a localized ridge all around the circumference) and all data points but one— corresponding to the maximum imperfection amplitude of 73% of the wall thickness—are bounded from below by the knockdown curve. Indeed, as demonstrated by Koiter [10], axisymmetric sinusoidal imperfections cause the greatest erosion in buckling load. For these particular imperfections, and with pronounced imperfection amplitudes, the cylinder could buckle below the dashed curved of figure 10b. However, as pointed out by Eßlinger & Geier [12], axisymmetric undulations represent engineered (structured) imperfections that are unlikely to

occur naturally. Furthermore, this scenario seems increasingly unlikely for cylinders of the modern age, which can be manufactured to geometric tolerances smaller than the wall thickness. Hence, coupled to the knockdown factor proposed in figure 10*b* comes the understanding that imperfection control using modern engineering standards is being imposed.

One difference between NASA's empirical lower-bound curve (SP-8007) and the dashed curve of figure 10*b* is that the former is formulated as a function of $R/t$, and the latter as a function of $L^2/Rt$. This implies that the SP-8007 threshold and the present shock-sensitivity threshold can not be plotted as single curves on the same set of axes. However, many experimental results from the early 20th century that were used to formulate SP-8007 (e.g. [52]) fall below the dashed curve of figure 10*b*. Furthermore, while the SP-8007 guideline converges to $P/P_{cl} \approx 0.2$ for large $R/t$, the present shock-sensitivity threshold of figure 10*b* converges to $P/P_{cl} \approx 0.35$ for large $L^2/Rt$. It is interesting to note that in a recent publication [53] on imperfect cylinders, the knockdown factor approaches a similar value of $P/P_{cl} \approx 0.35$ with increasing dimple imperfection amplitude (normalized by cylinder thickness). Furthermore, this trend persists for different $R/t$ and $L/R$ ratios (assuming the cylinder is not very short), providing further credence to the newly developed knockdown factor. The more conservative nature of NASA's SP-8007 guideline can be attributed to the fact that many of the studies included in SP-8007 relied on manufacturing technology much less developed than today's and idealized boundary conditions could not be replicated as faithfully in experiments. If the geometry and test set-up are closely controlled, the onset of shock sensitivity as denoted by equation (7.1), may provide a less conservative (safe) design load.

## 8. Conclusion

Throughout this paper, we have addressed multiple aspects pertaining to the role of localized post-buckling states in axially compressed cylinders. Using nonlinear quasi-static finite-element methods and numerical continuation algorithms we traced the evolution of a single dimple into an axially localized/circumferentially periodic ring of diamond buckles. The pattern formation occurs via a homoclinic snaking sequence that describes a series of odd buckles. A second intertwined snaking sequence of even buckles starting from two dimples also exists, completing the typical picture of homoclinic snaking. The single ring of axially localized/circumferentially periodic diamond buckles destabilizes at a pitchfork bifurcation, initiating the formation of additional rings of buckles. Rather than snapping into existence in their entirety, these rings also form sequentially via circumferential growth starting from a single dimple. Hence, circumferential snaking seems to act as an organizing centre for the pattern formation in axial compressed cylinders and this is supported in the literature by high-speed photography of experiments.

The stability landscape to lateral probing reveals that the single-dimple forms a decreasing energy barrier between the pre-buckling and post-buckling regimes. In fact, the unstable ridge that bounds the basin of attraction of the pre-buckling well shows a whole plethora of localizations with low energy barriers. Due to the rotational invariance of the cylinder, these localizations can appear anywhere around the circumference of the shell and this multi-valued nature means that there is a large set of edge states that can be eroded by initial imperfections or traversed by disturbances. How and which energy barrier is eroded in an experiment is thus strongly dependent on the precise nature of initial and evolving testing conditions.

Indeed, as shown in previous research [4], the single-dimple localization forms the smallest energy barrier—a so-called mountain-pass point—between the stable pre-buckling well and a regime of restabilized post-buckling states of lower total potential energy. As shown here, the magnitude of the separating energy barrier is at most 0.4% greater than the energy in the pre-buckling state. Thus, once the single dimple exists as a saddle point, the cylinder can readily transition out of the pre-buckling energy well through the influence of external disturbances, perturbations, or through the eroding effects of initial imperfections; i.e. the cylinder can be described as in a state of heightened 'shock sensitivity' [43].

In this sense, the compressive onset of shock sensitivity could serve as an estimate for the buckling load of imperfect cylinders. The level of end-shortening for which the single-dimple localization exists as a saddle is denoted by a limit point. Tracing the critical boundary of this limit point with respect to geometric parameters (radius $R$, thickness $t$ and length $L$) shows that the knockdown factor associated with the onset of shock sensitivity is approximately proportional to the non-dimensional parameter $\eta = R^{0.16}t^{0.16}L^{-0.32}$. Comparisons with more than 100 experimental data points from the literature suggests that a buckling load estimate based on this parameter bounds the experimental data on carefully manufactured cylinders from below, and provides a less conservative design load than NASA's SP-8007 guideline [9].

The notions of localizations and point-wise disturbances reflects the hypothesis formulated by Eßlinger & Geier [12] that localized stress fields trigger premature buckling. If this is indeed the case, then tailoring stress fields in a particular (non-uniform) way should result in better correlations between predictions and experiments; the reasoning being that the symmetry of a circumferentially uniform stress field can be broken by a larger set of imperfections than an *a priori* tailored, non-uniform stress field. This could explain the recent success of researchers in deterministically tailoring and predicting the precise post-buckling behaviour of shells with engineered imperfections [8,54] and also with varying material properties [55]. Especially, the latter work may provide an impetus for future work on tailoring cylinders with modern materials technology, such as tow-steered composites.

Data accessibility. The underlying data to reproduce all graphs are available at the University of Bristol data repository, data.bris, at https://doi.org/10.5523/bris.9s3o74cpno1g2hivpnc0sjcod.

Authors' contributions. R.M.J.G. and A.P. conceived the work. R.M.J.G. developed the numerical tools and carried out the simulations in consultation with A.P. Both authors interpreted the results, wrote the paper and gave final approval for publication.

Competing interests. We declare we have no competing interests.

Funding. This work was supported by the Royal Academy of Engineering under the Research Fellowship scheme (grant no. RF\201718\17178) and the UK Engineering and Physical Sciences Research Council (grant no. EP/M013170/1).

Acknowledgements. The authors thank Prof. G. W. Hunt for insightful discussions about the Maxwell energy criterion and its relation to the snaking sequences described herein.

# Appendix A. Deformation modes of additional states

Figure 11 shows the additional deformation modes of the cylinder as referred to in §§4b and 5, respectively.

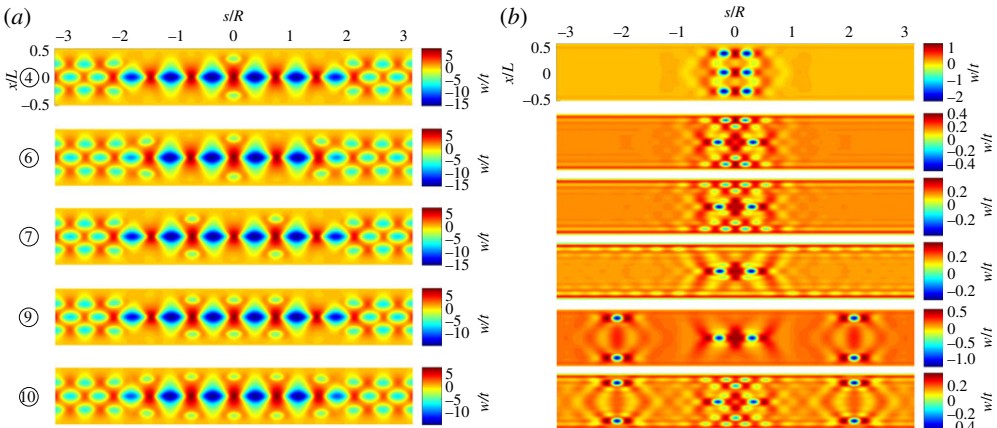

**Figure 11.** Additional deformation modes of the cylinder corresponding to: (*a*) the deep post-buckled behaviour of figure 5*a* and (*b*) the localized states of figure 8*a*. (Online version in colour.)

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
