## [Reviewer comments · Proceedings. Mathematical, Physical, and Engineering Sciences]

Review History

RSPA-2019-0006.R0 (Original submission)

Review form: Referee 1

Is the manuscript an original and important contribution to its field?

Yes

Is the paper of sufficient general interest?

Yes

Is the overall quality of the paper suitable?

Yes

Quality of the paper

An outstanding paper of the highest international importance; a major contribution to the field: must be published.

Can the paper be shortened without overall detriment to the main message?

No

Do you think some of the material would be more appropriate as an electronic appendix?

No

For papers with colour figures – is colour essential?

Yes

If there is supplementary material, is this adequate and clear?

Not applicable

Are there details of how to obtain materials and data, including any restrictions that may apply?

Not applicable

Do you have any ethical concerns with this paper?

No

Recommendation?

Accept as is

Comments to the Author(s)

Please see the attached pdf review file

Review form: Referee 2 (Giles Hunt)

Is the manuscript an original and important contribution to its field?

Yes

Is the paper of sufficient general interest?

Yes

Is the overall quality of the paper suitable?

Yes

Quality of the paper

An excellent paper making an important contribution to the field: should be published.

Can the paper be shortened without overall detriment to the main message?

No

Do you think some of the material would be more appropriate as an electronic appendix?

No

For papers with colour figures – is colour essential?

Yes

If there is supplementary material, is this adequate and clear?

Not applicable

Are there details of how to obtain materials and data, including any restrictions that may apply?

Yes

Do you have any ethical concerns with this paper?

No

Recommendation?

Accept with minor revision (please list in comments)

Comments to the Author(s)

I think this is an excellent paper well worthy of publication in Proc Roy Soc A. It follows a succession of recent papers focussing on a process of instability in thin axially-loaded cylindrical shells that identify a clear-cut triggering mechanism involving initiation from a single dimple indent on the shell surface. Not only is this the most likely defect to occur in practice, but under certain circumstances it is all that would be required to produce failure over the whole shell. The paper certainly provides a fresh in-depth study of the cylinder problem, and represents a significant extra step on a much-studied system. I feel however that some additional interpretations/commentary would be useful and shall be drawing attention to various instances where these could be provided in what follows.

The paper differs from the earlier work in that, rather being derived from underlying shell (von-Karman-Donnell) equations, it is produced from a formulation employing isoparametric, geometrically-nonlinear finite shell elements. One advantage of this is that it is able to model realistic experimental boundary conditions, with the effect that the nonlinear pre-buckling behaviour of an axially-symmetric ring buckling mode spreading from the ends is reproduced. It is interesting to see that this state is disconnected from the single dimple buckling solution, drawn attention to in Figs. 3 and 4, a distinction that I think must be lost in the earlier formulations where the boundary conditions allow the same linear constant-dilation pre-buckled state at every point on the shell. This brings additional significant numerical challenges. Whether this disconnection carries any practical significance for long shells is to my mind an open question, but it does raise some interesting issues worthy of further reflection.

It is interesting that the FE model seems to want to buckle with $n = 10$ or 9 circumferential waves once the dimple has propagated into circumferential periodicity, with the 9 -form being not quite periodic. This is a slight mismatch with the actual Yamaki experiment, which buckles initially into $n = 11$ and find $n = 10$ then 9 later in the evolution. As buckling progresses axially this wavenumber drops, so it's a bit surprising that the computed wavenumber starts at a lower value than found experimentally. I wonder whether this has something to do to the constraint of imposing symmetry on the quarter shell, but as the authors simply mention only "pertinent" boundary conditions it's hard to know. Odd wavenumbers, for instance which might naturally ask for symmetry at one boundary and anti-symmetry at another, could perhaps be being denied. If so, this is a pity and could of course be overcome by computations over the complete shell as the authors suggest for future work. It will be interesting to see whether this is indeed the case. I suggest that the symmetry conditions should perhaps be more carefully described, and some comment on the constraints of symmetry assumptions be added, possibly in the concluding remarks.

The paper seems to suggest that the SP-8007 recommendation coincides with the lowest load that can sustain an inwards dimple induced with a probe, which makes sense but also as the authors suggest is likely to be ultra-conservative. They suggest that a more realistic value may be obtained from the position of the single dimple limit point marked as SD, which they also discuss in connection with the Maxwell displacement for a periodic ring of 10 buckles. But this doesn't

take into account the domino effect of a single dimple triggering immediately a complete ring of buckles, which can occur at constant end-displacement without any extra input of energy over and above that of a single shock, but only above the Maxwell displacement. This to my mind is likely to be a highly significant mechanism for complete collapse, and I think puts new emphasis on the Maxwell displacement for the single dimple. Perhaps this whole issue could do with a little extra discussion in the final draft.

A few points relating to specific places in the paper follow:

Page 5 line 11. I have never really felt that the Esslinger and Geier experiments provided indisputable evidence of fully localized buckling. As I understand it (the German-speaking first author may be able to confirm this) they were performed with the heating effect of the lighting system providing the trigger for the buckling process. Among other things, this ensured that the initial buckle occurred in the visible region rather than at the back of the sample. This of course doesn't mean that the process wouldn't start locally under distributed loading conditions, only that the Esslinger and Geier experiments do not necessarily provide supporting evidence for this.

Page 5 line 37. "de- a re-stabilisations"?? There seems some slight typo here.

Page 9 line 20. Reference 20 also seems pertinent here. There is also a second paper by the same authors which, I think for the first time, refers to the relevance of the Maxwell criterion to localization and, by inference, to snaking.

Page 10 line 47. I find the appearance of the so-called "smaller" (secondary) snaking phenomena most fascinating and note these they do not seem to appear in the earlier work based on the von Karman-Donnell equations (refs 4 and 23 for example). I had wondered whether this might have been a result of the discretization, but it seems not. The number of times it occurs seems to be linked to the number of full dimples that have already formed, but beyond that it's hard to visualize a convincing mechanism for this effect. I guess it must remain a subject for further research, and to this end it might be instructive to see representations of the deflected shape at points within the secondary snake. Perhaps, after reflection, the authors might have some further comments to make on this issue.

To conclude, I repeat that I think this is an excellent contribution. The above suggestions are not intended as criticism, but are merely given in the hope that the final version may be yet further improved.

Giles Hunt, Emeritus Professor, University of Bath

Decision letter (RSPA-2019-0006.R0)

27-Feb-2019

Dear Dr Groh,

On behalf of the Editor, I am pleased to inform you that your Manuscript RSPA-2019-0006 entitled "On the role of localisations in buckling of axially compressed cylinders" has been accepted for publication subject to minor revisions in Proceedings A. Please find the referees' comments below.

The reviewer(s) have recommended publication, but also suggest some minor revisions to your manuscript. Therefore, I invite you to respond to the reviewer(s)' comments and revise your manuscript. Please note that we have a strict upper limit of 28 pages for each paper. Please endeavour to incorporate any revisions while keeping the paper within journal limits. Please note that page charges are made on all papers longer than 20 pages. If you cannot pay these charges you must reduce your paper to 20 pages before submitting your revision. Your paper has been ESTIMATED to be 27 pages. We cannot proceed with typesetting your paper without your agreement to meet page charges in full should the paper exceed 20 pages when typeset. If you have any questions, please do get in touch.

It is a condition of publication that you submit the revised version of your manuscript within 7 days. If you do not think you will be able to meet this date please let me know in advance of the due date.

To revise your manuscript, log into <https://mc.manuscriptcentral.com/prsa> and enter your Author Centre, where you will find your manuscript title listed under "Manuscripts with Decisions." Under "Actions," click on "Create a Revision." Your manuscript number has been appended to denote a revision.

You will be unable to make your revisions on the originally submitted version of the manuscript. Instead, revise your manuscript and upload a new version through your Author Centre.

IMPORTANT: Your original files are available to you when you upload your revised manuscript. Please delete any redundant files before completing the submission process.

In addition to addressing all of the reviewers' and editor's comments, your revised manuscript **MUST** contain the following sections before the reference list (for any heading that does not apply to your work, please include a comment to this effect):

- Acknowledgements
- Funding statement

See <https://royalsociety.org/journals/authors/author-guidelines/> for further details.

When uploading your revised files, please make sure that you include the following as we cannot proceed without these:

- 1) A text file of the manuscript (doc, txt, rtf or tex), including the references, tables (including captions) and figure captions. Please remove any tracked changes from the text before submission. PDF files are not an accepted format for the "Main Document".
- 2) A separate electronic file of each figure (tif, eps or print-quality pdf preferred). The format should be produced directly from original creation package, or original software format.
- 3) Electronic Supplementary Material (ESM): all supplementary materials accompanying an accepted article will be treated as in their final form. Note that the Royal Society will not edit or

typeset supplementary material and it will be hosted as provided. Please ensure that the supplementary material includes the paper details where possible (authors, article title, journal name). Supplementary files will be published alongside the paper on the journal website and posted on the online figshare repository (<https://figshare.com>). The heading and legend provided for each supplementary file during the submission process will be used to create the figshare page, so please ensure these are accurate and informative so that your files can be found in searches. Files on figshare will be made available approximately one week before the accompanying article so that the supplementary material can be attributed a unique DOI. Alternatively you may upload a zip folder containing all source files for your manuscript as described above with a PDF as your "Main Document". This should be the full paper as it appears when compiled from the individual files supplied in the zip folder.

Article Funder

Please ensure you fill in the Article Funder question on page 2 to ensure the correct data is collected for FundRef (<http://www.crossref.org/fundref/>).

Media summary

Please ensure you include a short non-technical summary (up to 100 words) of the key findings/importance of your paper. This will be used for to promote your work and marketing purposes (e.g. press releases). The summary should be prepared using the following guidelines:

*Write simple English: this is intended for the general public. Please explain any essential technical terms in a short and simple manner.

*Describe (a) the study (b) its key findings and (c) its implications.

*State why this work is newsworthy, be concise and do not overstate (true 'breakthroughs' are a rarity).

*Ensure that you include valid contact details for the lead author (institutional address, email address, telephone number).

Cover images

We welcome submissions of images for possible use on the cover of Proceedings A. Images should be square in dimension and please ensure that you obtain all relevant copyright permissions before submitting the image to us. If you would like to submit an image for consideration please send your image to proceedingsa@royalsociety.org

Once again, thank you for submitting your manuscript to Proceedings A and I look forward to receiving your revision. If you have any questions at all, please do not hesitate to get in touch.

Best wishes

Alice Power
Publishing Editor
Proceedings A
proceedingsa@royalsociety.org

on behalf of
Dr Tim Dodwell
Board Member
Proceedings A

Reviewer(s)' Comments to Author:

Referee: 1

Comments to the Author(s)

please see the attached pdf review file

Referee: 2

Comments to the Author(s)

I think this is an excellent paper well worthy of publication in Proc Roy Soc A. It follows a succession of recent papers focussing on a process of instability in thin axially-loaded cylindrical shells that identify a clear-cut triggering mechanism involving initiation from a single dimple indent on the shell surface. Not only is this the most likely defect to occur in practice, but under certain circumstances it is all that would be required to produce failure over the whole shell. The paper certainly provides a fresh in-depth study of the cylinder problem, and represents a significant extra step on a much-studied system. I feel however that some additional interpretations/commentary would be useful and shall be drawing attention to various instances where these could be provided in what follows.

The paper differs from the earlier work in that, rather being derived from underlying shell (von-Karman-Donnell) equations, it is produced from a formulation employing isoparametric, geometrically-nonlinear finite shell elements. One advantage of this is that it is able to model realistic experimental boundary conditions, with the effect that the nonlinear pre-buckling behaviour of an axially-symmetric ring buckling mode spreading from the ends is reproduced. It is interesting to see that this state is disconnected from the single dimple buckling solution, drawn attention to in Figs. 3 and 4, a distinction that I think must be lost in the earlier formulations where the boundary conditions allow the same linear constant-dilation pre-buckled state at every point on the shell. This brings additional significant numerical challenges. Whether this disconnection carries any practical significance for long shells is to my mind an open question, but it does raise some interesting issues worthy of further reflection.

It is interesting that the FE model seems to want to buckle with $n = 10$ or 9 circumferential waves once the dimple has propagated into circumferential periodicity, with the 9 -form being not quite periodic. This is a slight mismatch with the actual Yamaki experiment, which buckles initially into $n = 11$ and find $n = 10$ then 9 later in the evolution. As buckling progresses axially this wavenumber drops, so it's a bit surprising that the computed wavenumber starts at a lower value than found experimentally. I wonder whether this has something to do to the constraint of imposing symmetry on the quarter shell, but as the authors simply mention only "pertinent" boundary conditions it's hard to know. Odd wavenumbers, for instance which might naturally ask for symmetry at one boundary and anti-symmetry at another, could perhaps be being denied. If so, this is a pity and could of course be overcome by computations over the complete shell as the authors suggest for future work. It will be interesting to see whether this is indeed the case. I suggest that the symmetry conditions should perhaps be more carefully described, and some comment on the constraints of symmetry assumptions be added, possibly in the concluding remarks.

The paper seems to suggest that the SP-8007 recommendation coincides with the lowest load that can sustain an inwards dimple induced with a probe, which makes sense but also as the authors suggest is likely to be ultra-conservative. They suggest that a more realistic value may be obtained from the position of the single dimple limit point marked as SD, which they also discuss

in connection with the Maxwell displacement for a periodic ring of 10 buckles. But this doesn't take into account the domino effect of a single dimple triggering immediately a complete ring of buckles, which can occur at constant end-displacement without any extra input of energy over and above that of a single shock, but only above the Maxwell displacement. This to my mind is likely to be a highly significant mechanism for complete collapse, and I think puts new emphasis on the Maxwell displacement for the single dimple. Perhaps this whole issue could do with a little extra discussion in the final draft.

A few points relating to specific places in the paper follow:

Page 5 line 11. I have never really felt that the Esslinger and Geier experiments provided indisputable evidence of fully localized buckling. As I understand it (the German-speaking first author may be able to confirm this) they were performed with the heating effect of the lighting system providing the trigger for the buckling process. Among other things, this ensured that the initial buckle occurred in the visible region rather than at the back of the sample. This of course doesn't mean that the process wouldn't start locally under distributed loading conditions, only that the Esslinger and Geier experiments do not necessarily provide supporting evidence for this.

Page 5 line 37. "de- a re-stabilisations"?? There seems some slight typo here.

Page 9 line 20. Reference 20 also seems pertinent here. There is also a second paper by the same authors which, I think for the first time, refers to the relevance of the Maxwell criterion to localization and, by inference, to snaking.

Page 10 line 47. I find the appearance of the so-called "smaller" (secondary) snaking phenomena most fascinating and note these they do not seem to appear in the earlier work based on the von Karman-Donnell equations (refs 4 and 23 for example). I had wondered whether this might have been a result of the discretization, but it seems not. The number of times it occurs seems to be linked to the number of full dimples that have already formed, but beyond that it's hard to visualize a convincing mechanism for this effect. I guess it must remain a subject for further research, and to this end it might be instructive to see representations of the deflected shape at points within the secondary snake. Perhaps, after reflection, the authors might have some further comments to make on this issue.

To conclude, I repeat that I think this is an excellent contribution. The above suggestions are not intended as criticism, but are merely given in the hope that the final version may be yet further improved.

Giles Hunt, Emeritus Professor, University of Bath

Author's Response to Decision Letter for (RSPA-2019-0006.R0)

See Appendix A.

Decision letter (RSPA-2019-0006.R1)

07-Mar-2019

Dear Dr Groh

I am pleased to inform you that your manuscript entitled "On the role of localisations in buckling of axially compressed cylinders" has been accepted in its final form for publication in Proceedings A.

Our Production Office will be in contact with you in due course. You can expect to receive a proof of your article soon. Please contact the office to let us know if you are likely to be away from e-mail in the near future. If you do not notify us and comments are not received within 5 days of sending the proof, we may publish the paper as it stands.

Your article has been estimated as being 28 pages long. Our Production Office will inform you of the exact length at the proof stage.

Proceedings A levies charges for articles which exceed 20 printed pages. (based upon approximately 540 words or 2 figures per page). Articles exceeding this limit will incur page charges of £150 per page or part page, plus VAT (where applicable).

We are keen to promote all published material in the journal. If you wish us to highlight the publication of your paper to appropriate colleagues, please send me by return email the names and email addresses of up to 5 people and we will ensure that they are notified once the paper goes online.

Under the terms of our licence to publish you may post the author generated postprint (ie. your accepted version not the final typeset version) of your manuscript at any time and this can be made freely available. Postprints can be deposited on a personal or institutional website, or a recognised server/repository. Please note however, that the reporting of postprints is subject to a media embargo, and that the status the manuscript should be made clear. Upon publication of the definitive version on the publisher's site, full details and a link should be added.

You can cite the article in advance of publication using its DOI. The DOI will take the form: 10.1098/rspa.XXXX.YYYY, where XXXX and YYYY are the last 8 digits of your manuscript number (eg. if your manuscript number is RSPA-2017-1234 the DOI would be 10.1098/rspa.2017.1234).

For tips on promoting your accepted paper see our blog post:
<https://blogs.royalsociety.org/publishing/promoting-your-latest-paper-and-tracking-your-results/>

On behalf of the Editor of Proceedings A, we look forward to your continued contributions to the Journal.

Sincerely,

Alice Power
Publishing Editor
Proceedings A
proceedingsa@royalsociety.org

Appendix A

The authors would like to thank the referees for their thoughtful and careful reading of the manuscript. The referees' input to improve the quality of the paper is much appreciated. The comments made by both referees are reproduced below, and where appropriate, the authors' response is provided immediately beneath. Additions and changes to the original manuscript are highlighted in red.

Referee #1:

1) *This is an impressive paper. It definitely advances the foundations of buckling theory generally and more specifically our understanding of the elastic buckling of cylindrical shells. The paper builds on and comprehensively advances recent new mathematical approaches to shell buckling. For a difficult subject, I find the paper to be well written and readable. I'll go farther by saying that for a paper making use of these new (non-trivial) mathematical methods the authors display an impressive command and knowledge of the physical aspects of cylindrical shell buckling. The culmination of the work leading to Fig. 10b is particularly important and insightful. I enthusiastically recommend publication with one very minor suggestion and one observation making connection to the authors' efforts to identify a effective 'lower bound' buckling criterion. Both points are optional as far as publication is concerned.*

Response: Thank you for these positive comments. We have incorporated both suggestions raised by the referee as outlined further below.

2) *The background discussion in the introduction is excellent. In the last half of the introduction, the authors give extensive discussion making the tacit assumption that the shell is loaded under what they call rigid displacement loading. I think the reader should be informed of this tacit assumption. For example, if the shell were loaded with prescribed force (dead load) much of what is discussed in the last half of the introduction is irrelevant—because once over the mountain pass the shell undergoes complete collapse.*

Response: This is an excellent point. The pattern formation triggered by traversal over the mountain pass only occurs for displacement-controlled (rigid) loading. To make the current tacit assumption more explicit, we have added the following sentence at the beginning of the second half of the introduction:

Interestingly, for a specific level of end-shortening, the single dimple identified by Horák *et al.* [4] corresponds to the smallest energy barrier between the pre-buckled state and a restabilised post-buckling state of lower energy. **An important caveat is that the cylinder only restabilises under controlled end-shortening (rigid loading), whereas in the case of force-controlled loading (dead loading), the cylinder undergoes complete collapse once the mountain pass has been crossed. With this caveat in mind, consider,** for example, the two energy wells depicted in Figure 1b, with one valley representing a high-energy pre-buckling equilibrium and the other a restabilised, low-energy post-buckling equilibrium (of localised or periodic form).

3) *The findings in this paper align quite nicely with those in a recently published paper (S. Gerasimidis, et al., On Establishing Buckling Knockdowns... J. Appl. Mech. (2018) 85, 091010-1-14). I include below Fig. 14 from this paper which presents the range of P/PC for cylindrical shells based on Koiter's worse case imperfection and new results based on more realistic dimple imperfections. Note that for the largest imperfection amplitude shown, the buckling load for the dimple imperfections is about 0.35 PC in accord with estimates found by the authors based on their results for the perfect shell. It is important to note that the dimensionless results in Fig. 14 below are independent of both R/t and L/R , assuming the shell is not very short. Thus, this reviewer is quite certain that shells with realistic imperfections can buckle below the authors' curve in Fig 10b, as the one outlier illustrates, but it does require sufficiently large imperfection amplitudes. As the authors tacitly note, coupled to a criterion such as the one they*

are proposing in Fig. 10b is the understanding that some level of imperfection control must be imposed. The outlier and Fig. 14 make that clear.

Response: Thank you for pointing out this pertinent recent publication. This paper had slipped under our radar, but we are pleased to see the asymptotic knockdown factor of 0.35 appear in the related context of dimple imperfection amplitude. We have added the following short discussion beneath Figure 10:

It is interesting to note that in a recent publication [53] on imperfect cylinders, the knockdown factor approaches a similar value of $P/P_{cl} \sim 0.35$ with increasing dimple imperfection amplitude (normalised by cylinder thickness). Furthermore, this trend persists for different R/t and L/R ratios (assuming the cylinder is not very short), providing further credence to the newly developed knockdown factor.

With regards to Koiter's axisymmetric sinusoidal imperfection, the following section below Figure 10 has been rewritten as follows:

Indeed, as demonstrated by Koiter [10], axisymmetric sinusoidal imperfections cause the greatest erosion in buckling load. For these particular imperfections, and with pronounced imperfection amplitudes, the cylinder could buckle below the dashed curved of Figure 10b. However, as pointed out by Eßlinger & Geier [12], axisymmetric perturbations represent engineered (structured) imperfections that are unlikely to occur naturally. Furthermore, this scenario seems increasingly unlikely for cylinders of the modern age, which can be manufactured to geometric tolerances smaller than the wall thickness. Hence, coupled to the knockdown factor proposed in Figure 10b comes the understanding that imperfection control using modern engineering standards is being imposed.

References Added

The following reference has been added to the manuscript:

- S. Gerasimidis, E. Virost, J. W. Hutchinson, S. M. Rubinstein. On Establishing Buckling Knockdowns for Imperfection-Sensitive Shell Structures. Journal of Applied Mechanics, 85:091010, 2018.

Referee #2:

1) I think this is an excellent paper well worthy of publication in Proc Roy Soc A. It follows a succession of recent papers focussing on a process of instability in thin axially-loaded cylindrical shells that identify a clear-cut triggering mechanism involving initiation from a single dimple indent on the shell surface. Not only is this the most likely defect to occur in practice, but under certain circumstances it is all that would be required to produce failure over the whole shell. The paper certainly provides a fresh in-depth study of the cylinder problem, and represents a significant extra step on a much-studied system. I feel however that some additional interpretations/commentary would be useful and shall be drawing attention to various instances where these could be provided in what follows.

Response: Thank you for your positive comments. Responses to individual suggestions are provided below.

2) The paper differs from the earlier work in that, rather being derived from underlying shell (von-Karman-Donnell) equations, it is produced from a formulation employing isoparametric, geometrically-nonlinear finite shell elements. One advantage of this is that it is able to model realistic experimental

boundary conditions, with the effect that the nonlinear pre-buckling behaviour of an axially-symmetric ring buckling mode spreading from the ends is reproduced. It is interesting to see that this state is disconnected from the single dimple buckling solution, drawn attention to in Figs. 3 and 4, a distinction that I think must be lost in the earlier formulations where the boundary conditions allow the same linear constant-dilation pre-buckled state at every point on the shell. This brings additional significant numerical challenges. Whether this disconnection carries any practical significance for long shells is to my mind an open question, but it does raise some interesting issues worthy of further reflection.

Response: It was equally surprising to us that the single-dimple localisation should be broken away from the nonlinear pre-buckling path. As pointed out by the referee, compared to previous findings in the literature with linear pre-buckling, this may just be an artefact of the nonlinear pre-buckling path. Another hypothesis is that the imposed symmetry conditions lead to constraints that force this feature to occur. We believe this scenario to be unlikely. Rather it is possible that the limit point that delimits the broken-away portion of the single-dimple path occurs as a broken-away or perfect bifurcation of a periodic buckling path emanating from the pre-buckling path. This possibility was raised in the text and is a real possibility as (a) there are many periodic post-buckling paths that lead to localisations that were found, but not plotted in the paper (for conciseness); and (b) branching from periodic to localised post-buckling paths occurs in other subcritical systems. The nature of the broken-away path and potential influence of symmetry boundary conditions will be studied further in future work on the full cylinder, as further elucidated in additional responses below.

3) It is interesting that the FE model seems to want to buckle with $n = 10$ or 9 circumferential waves once the dimple has propagated into circumferential periodicity, with the 9-form being not quite periodic. This is a slight mismatch with the actual Yamaki experiment, which buckles initially into $n = 11$ and find $n = 10$ then 9 later in the evolution. As buckling progresses axially this wavenumber drops, so it's a bit surprising that the computed wavenumber starts at a lower value than found experimentally. I wonder whether this has something to do to the constraint of imposing symmetry on the quarter shell, but as the authors simply mention only "pertinent" boundary conditions it's hard to know. Odd wavenumbers, for instance which might naturally ask for symmetry at one boundary and anti-symmetry at another, could perhaps be being denied. If so, this is a pity and could of course be overcome by computations over the complete shell as the authors suggest for future work. It will be interesting to see whether this is indeed the case. I suggest that the symmetry conditions should perhaps be more carefully described, and some comment on the constraints of symmetry assumptions be added, possibly in the concluding remarks.

Response: A comment to describe the symmetry conditions more clearly is added in Section 2:

To reduce the computational effort and complexity of the problem, only a quarter of the cylinder is modelled with the pertinent **mirror** symmetry conditions applied at the cylinder half-length and half-circumference. **The imposed translational and rotational symmetry conditions prevent movement of boundary nodes across the symmetry plane and constrain rotations around the line of symmetry.**

With regards to the buckling mode progression, we would like to clarify a point that, perhaps, was not made explicitly clear in the manuscript. Two different snaking mechanisms are described, one with odd number of buckles and the second with even number of buckles, both of which lead to the periodic state of $n = 10$ buckles. A mode shape of 9 buckles is shown in the sequence of odd buckles, but this is only done to show the snaking progression, and it is unlikely that the cylinder would ever

stabilise in this mode. Furthermore, this mode is not the 9 buckle mode shown by Yamaki as it is not periodic, and therefore an entirely different mode than the one observed by Yamaki. Hence, the aperiodic 9 buckle mode should be viewed as a transitional mode to the periodic mode of 10 buckles, where the latter is the mode shape we suggest the cylinder would stabilise in. This, however, leaves the question of why the buckling mode shape does not correspond to Yamaki's 11 buckle mode. It is entirely possible that the 11 buckle mode exists as a solution in the equilibrium manifold, but our results (and indeed the results by Kreilos & Schneider) suggest that this 11 buckle mode cannot exist on one continuous equilibrium path starting from a single dimple. In the naturally dynamical world of an actual experiment, the cylinder would, nevertheless, still be free to transition and stabilise in this mode once instability first arises. To address this point, we have added the following explanation at the end of Section 4a:

Finally, we note that Yamaki's experiments on the present cylinder [28] showed initial buckling into an eleven-form buckling mode that then transitioned to ten and later to nine buckles as end-shortening was increased. It is possible that the buckling mode with eleven waves exists as a separate equilibrium path on the response diagram, but our analyses, and indeed the analyses by Kreilos & Schneider [23], suggest that it is the ten-form mode that lies on one continuous path starting from the single dimple. In the dynamic domain of an actual experiment, the cylinder could, nevertheless, transition and stabilise into this alternative mode of eleven buckles once the first instability arises. Another possibility is that the imposed symmetry boundary conditions prevent bifurcations that lead to a buckling mode with eleven-fold periodicity. This possibility provides further motivation to extend the analysis to the full cylinder.

4) *The paper seems to suggest that the SP-8007 recommendation coincides with the lowest load that can sustain an inwards dimple induced with a probe, which makes sense but also as the authors suggest is likely to be ultra-conservative. They suggest that a more realistic value may be obtained from the position of the single dimple limit point marked as SD, which they also discuss in connection with the Maxwell displacement for a periodic ring of 10 buckles. But this doesn't take into account the domino effect of a single dimple triggering immediately a complete ring of buckles, which can occur at constant end-displacement without any extra input of energy over and above that of a single shock, but only above the Maxwell displacement. This to my mind is likely to be a highly significant mechanism for complete collapse, and I think puts new emphasis on the Maxwell displacement for the single dimple. Perhaps this whole issue could do with a little extra discussion in the final draft.*

Response: The SP-8007 recommendation does not coincide with the lowest load that can sustain an inwards dimple induced with a probe. Rather, as mentioned in Section 1, it is a statistical lower fit to experimental data on cylinders tested prior to 1960. The reason this guideline has proven itself to be too conservative for modern cylinders is that manufacturing tolerances have improved significantly over the last 60 years, and hence cylinders of much better quality can be manufactured.

The connection between the Maxwell criterion and limit point SD is discussed in Section 6, and in the present case, the Maxwell displacement is almost identical to limit point SD. We entirely agree with the referee's comment that triggering the single dimple can lead to a domino effect and the development of the full ring of buckles, and this mechanism is indeed shown in the discussion of the Maxwell criterion in the inset of Figure 9b. The reason we have chosen the limit point SD is two-fold. First, it corresponds closely to the Maxwell displacement, and second it denotes the onset of the single dimple as a mountain-pass point. For lower levels of end-shortening the single dimple does not exist as a mountain-pass point, and given the connection between shock sensitivity and the existence of a

mountain pass point, as outlined in Section 6, limit point SD seems a rational choice for a more deterministic knockdown design curve.

A few points relating to specific places in the paper follow:

5) Page 5 line 11. I have never really felt that the Esslinger and Geier experiments provided indisputable evidence of fully localized buckling. As I understand it (the German-speaking first author may be able to confirm this) they were performed with the heating effect of the lighting system providing the trigger for the buckling process. Among other things, this ensured that the initial buckle occurred in the visible region rather than at the back of the sample. This of course doesn't mean that the process wouldn't start locally under distributed loading conditions, only that the Esslinger and Geier experiments do not necessarily provide supporting evidence for this.

Response: As far as we are aware from the following two publications by Eßlinger, and Eßlinger & Geier, no heating was used to initiate buckling.

i) M. Eßlinger. Hochgeschwindigkeitsaufnahmen vom Beulvorgang dünnwandiger, axialbelasteter Zylinder. *Der Stahlbau*, 39(3):73–76, 1970.

ii) M. Eßlinger and B. Geier. Gerechnete Nachbeullasten als untere Grenze der experimentellen axialen Beullasten von Kreiszyklindern. *Der Stahlbau*, 41(12):353–360, 1972.

In the first publication, Eßlinger states that: “Die Zylinder wurden durch gesteuerte Stauchung belastet. Die Stauchungsgeschwindigkeit war so niedrig, dass das Auslösen des Beulvorgangs nicht durch dynamische Effekte beeinflusst wurde.” Which translates to: “The cylinders were loaded through controlled end-shortening. The compression velocity was so low that the initiation of buckling was not effected by dynamic effects.”

The second publication includes a picture of the test setup (shown below) with the following description. “Die Anlage ist deformationsgesteuert; der Vorschub wird mit Hilfe eines Exzentrers aufgebracht. Die verschiebbliche untere Platte ist parallel geführt.”. Which translates to: “The testing rig is displacement controlled; the compression rate is applied with the help of an eccentric sheave. The moveable bottom plate is guided parallel to the top.”

We found no mention of thermally activated buckling in these two papers. Furthermore, the separate experiments performed by Tennyson [13] cited in the paper confirm the onset of buckling as a “very localised phenomenon”.

6) Page 5 line 37. “de- a re-stabilisations”?? There seems some slight typo here.

Response: Thank you. This typo has been corrected.

7) Page 9 line 20. Reference 20 also seems pertinent here. There is also a second paper by the same authors which, I think for the first time, refers to the relevance of the Maxwell criterion to localization and, by inference, to snaking.

Response: Indeed, the paper mentioned by the referee is highly relevant to the discussion of the Maxwell criterion and has therefore been added in Section 6.

To address this problem of converging, yet non-intersecting equilibrium paths in the axially compressed cylinder, Hunt & Lucena Neto [42] adopt the Maxwell energy criterion.

8) Page 10 line 47. I find the appearance of the so-called “smaller” (secondary) snaking phenomena most fascinating and note these they do not seem to appear in the earlier work based on the von Karman-Donnell equations (refs 4 and 23 for example). I had wondered whether this might have been a result of the discretization, but it seems not. The number of times it occurs seems to be linked to the number of full dimples that have already formed, but beyond that it’s hard to visualize a convincing mechanism for this effect. I guess it must remain a subject for further research, and to this end it might be instructive to see representations of the deflected shape at points within the secondary snake. Perhaps, after reflection, the authors might have some further comments to make on this issue.

Response: The presence of these local snaking phenomena is indeed intriguing. A short comment was included in the original manuscript to this effect:

“Interestingly, smaller snaking phenomena also occur on the global snaking “fingers”, with the number of local turning points increasing with the number of buckles on each global snaking finger.”

It is indeed correct that this local snaking did not occur in the work based on von Karman-Donnell equations, but this might be explained by the differences in the kinematics (moderate rotations versus large rotation, total Lagrangian framework). The mode shapes of these local snakes were not included in the paper for conciseness and because the mode shapes did not change appreciably on these small segments. There are indeed small features that arise and then decay again as the two limit points of each secondary snake are passed, but these features are at least an order of magnitude smaller than the main buckles already present.

To provide some additional context the sentence cited above has been extended as follows:

Interestingly, smaller snaking phenomena also occur on the global snaking “fingers”, with the number of local turning points increasing with the number of buckles on each global snaking finger. These snaking phenomena were not observed in previous work based on Donnell-von Karman equations [23], and this might be explained by the different kinematics employed by the former moderate rotation and the current large rotation, total Lagrangian approach. The secondary snaking phenomena correspond to the initiation and subsequent decay of additional buckles, but as the corresponding deformations are at least an order of magnitude smaller than the existing buckles, they are unlikely to represent a significant aspect of the pattern formation.

To conclude, I repeat that I think this is an excellent contribution. The above suggestions are not intended as criticism, but are merely given in the hope that the final version may be yet further improved.

References Added

The following reference has been added to the manuscript:

- Hunt, G.W. and Lucena Neto, E. Maxwell Critical Loads for Axially Loaded Cylindrical Shells. Transactions of the American Society of Mechanical Engineers, 60:702-706, 1993.